# Bandits Meet Mechanism Design to Combat Clickbait in Online Recommendation

**Thomas Kleine Buening**[1], **Aadirupa Saha**[2]*, **Christos Dimitrakakis**[3], **Haifeng Xu**[4]

[1]The Alan Turing Institute, [2]TTIC, [3]University of Neuchatel, [4]University of Chicago

## Abstract

We study a strategic variant of the multi-armed bandit problem, which we coin the *strategic click-bandit*. This model is motivated by applications in online recommendation where the choice of recommended items depends on both the click-through rates and the post-click rewards. Like in classical bandits, rewards follow a fixed unknown distribution. However, we assume that the click-rate of each arm is chosen strategically by the arm (e.g., a host on Airbnb) in order to maximize the number of times it gets clicked. The algorithm designer does not know the post-click rewards nor the arms' actions (i.e., strategically chosen click-rates) in advance, and must learn both values over time. To solve this problem, we design an incentive-aware learning algorithm, UCB-S, which achieves two goals simultaneously: (a) incentivizing desirable arm behavior under uncertainty; (b) minimizing regret by learning unknown parameters. We approximately characterize all Nash equilibria of the arms under UCB-S and show a $\widetilde{\mathcal{O}}(\sqrt{KT})$ regret bound uniformly in *every* equilibrium. We also show that incentive-unaware algorithms generally fail to achieve low regret in the strategic click-bandit. Finally, we support our theoretical results by simulations of strategic arm behavior which confirm the effectiveness and robustness of our proposed incentive design.

## 1 Introduction

Recommendation platforms act as intermediaries between *vendors* and *users* so as to recommend *items* from the former to the latter. On Amazon, vendors sell physical items, while on Youtube the recommended items are videos. The recommendation problem is how to select one or more items to present to each user so that they are most likely to click on at least one of them.

However, vendor-chosen *item descriptions* are an essential aspect of the problem that is often ignored. These invite vendors to exaggerate their true value in the descriptions in order to increase their Click-Through-Rates (CTRs). As a consequence, even though online learning algorithms can generally identify relevant items, the existence of unrepresentative or exaggerated item descriptions remains a challenge (Yue et al., 2010; Hofmann et al., 2012). These include thumbnails or headlines that do not truly reflect the underlying item (see Figure 1)—a well-known internet phenomenon called the *clickbait* (Wang et al., 2021). While moderately increasing user click-rates through attractive descriptions is often encouraged since it helps to increase the overall user activity, clickbait can be harmful to a platform as it leads to bad recommendation outcomes and damage to the platform's reputation which may exceed the value of any additional clicks. A key reason for such dishonest or exaggerated item deceptions is the *strategic behavior* of vendors driven by their incentive to increase their item's exposure and click probability. Thus naturally, vendors are better off carefully choosing descriptions so as to increase click-rates, which leads to phenomena such as clickbait.[1]

To address this issue, we take an approach that marries *mechanism design* without payments with *online learning*, which are two celebrated research areas, however, mostly studied as separate streams. Since clickbait is fundamentally driven by vendor incentives, we believe that the novel design of online learning policies *that can carefully align vendor incentives with the platform's overall objective* may help to resolve this issue from its root.

---

*Author is currently with Apple ML Research.

[1]This is possible because most platforms rely on vendors to provide descriptions about their items. For instance, the images of restaurants on Yelp, rentals on Airbnb, hotels on Expedia, title and thumbnails of Youtube videos, and descriptions of products on Amazon are all provided by the vendors.

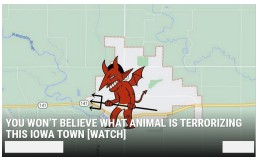 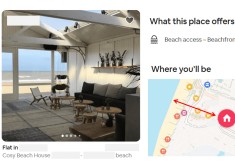 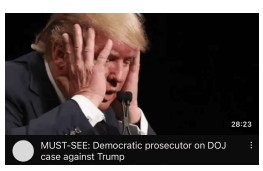 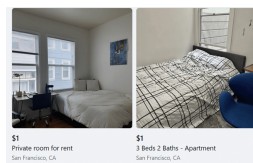

Figure 1: Examples of unrepresentative or clickbait headlines and thumbnails on Bing News, Airbnb, Youtube, and Facebook Marketplace (identifying information partly redacted).

To incorporate vendor-chosen item descriptions in this setting, we propose and study a natural strategic variant of the classical Multi-Armed Bandit (MAB) problem, which we call the *strategic click-bandit* in order to emphasize the strategic role that clicks and CTRs play in our setup.[2] Concretely, in strategic click-bandits, each arm $i$ is characterized by (a) a reward distribution with mean $\mu_i$, inherent to the arm; and (b) a click probability $s_i \in [0, 1]$, chosen freely by the arm at the beginning. Since the learner (i.e., the recommendation system) knows neither of these values in advance, it must learn them through interaction. The learner's objective is represented through a general utility function $u(s_i, \mu_i)$ that depends on both click-rate and post-click rewards.

We highlight two fundamental differences between strategic click-bandits and standard MABs. First, each arm in the strategic click-bandit is a *self-interested agent* whose objective is to maximize the number of times it gets clicked. This captures the strategic behavior of many vendors in online recommendations, especially those who are rewarded based on user clicks (e.g., Youtube (2023)). Second, $s_i$ is a freely chosen *action* by arm $i$, rather than a fixed parameter of arm $i$. We believe these modeling adjustments more realistically capture vendor behaviors in real applications. They also lead to intriguing mechanism design questions since the bandit algorithm not only needs to learn the unknown parameters, but also has to carefully align incentives to avoid undesired arm behavior. In summary, our contributions are:

1. We introduce the strategic click-bandit problem, which involves strategic arms manipulating click-rates so as to maximize their own utility, and show that *incentive-unaware* algorithms generally fail to achieve low regret in the strategic click-bandit (Section 3, Proposition 4.1).

2. We design an *incentive-aware* learning algorithm, UCB-S, that combines mechanism design and online learning techniques and effectively incentivizes desirable arm strategies while minimizing regret by making credible and justified threats to arms under uncertainty (Section 5).

3. We characterize the set of Nash equilibria for the arms under the UCB-S mechanism and show that every arm $i$'s strategy is $\tilde{\mathcal{O}}\big(\max\big\{\Delta_i, \sqrt{K/T}\big\}\big)$ close to the desired strategy in equilibrium (Theorem 5.2). We then show that UCB-S achieves $\tilde{\mathcal{O}}\big(\sqrt{KT}\big)$ strong strategic regret (Theorem 5.3) and complement this with an almost matching lower bound of $\Omega\big(\sqrt{KT}\big)$ for weak strategic regret (Theorem 5.5).

4. We simulate strategic arm behavior through repeated interaction and gradient ascent and empirically demonstrate the effectiveness of the proposed UCB-S mechanism (Section 6).

## 2 RELATED WORK

The MAB problem is a well-studied online learning framework, which can be used to model decision-making under uncertainty (Lai et al., 1985; Auer, 2002). Since it inherently involves sequential actions and the exploration-exploitation trade-off, the MAB framework has been applied to online recommendations (Li et al., 2010; Zong et al., 2016; Wang et al., 2017) as well as a myriad of other domains (Bouneffouf et al., 2020). While there is much work studying strategic machine learning (e.g., Hardt et al., 2016; Freeman et al., 2020; Zhang and Conitzer, 2021), we here wish to highlight related work that connects online learning (and specifically the MAB formalism) to mechanism design (Nisan and Ronen, 1999). Additional related work is discussed in Appendix H.

To the best of our knowledge, Braverman et al. (2019) are the first to study a strategic variant of the MAB problem. In their model, when an arm is pulled, it receives a privately observed reward $\nu$ and chooses to pass on a portion $x$ of it to the principal, keeping $\nu - x$ for itself. The goal of

---

[2]We use the terms click-through-rate, click-rate, and click probability interchangeably.

---

**Model 1:** The Strategic Click-Bandit Problem

---

1 Learner commits to algorithm $M$, which is shared with all arms
2 Arms choose strategies $(s_1, \ldots, s_K) \in [0, 1]^K$ (unknown to $M$)
3 **for** $t = 1, \ldots, T$ **do**
4      Algorithm $M$ selects arm $i_t \in [K]$
5      Arm $i_t$ is clicked with probability $s_{i_t}$, i.e., $c_{t,i_t} \sim \mathrm{Bern}(s_{i_t})$
6      **if** $i_t$ was clicked ($c_{t,i_t} = 1$) **then**
7          Arm $i_t$ receives utility 1 from the click
8          $M$ observes post-click reward $r_{t,i_t}$ drawn from a distribution with mean $\mu_{i_t}$

---

the principal is then to incentivize arms to share as much reward with the principal as possible. In contrast to our work, the principal must not learn the underlying reward distribution or the arm strategies, but instead design an auction among arms based on the shared rewards. Feng et al. (2020) and Dong et al. (2022) study the robustness of bandit algorithms to strategic reward manipulations. However, neither work attempts to align incentives by designing mechanisms, but instead assume a limited manipulation budget. Shin et al. (2022) study MABs with strategic replication in which agents can submit several arms with replicas to the platform. They design an algorithm, which separately explores the arms submitted by each agent and in doing so discourages agents from creating additional arms and replicas. Another line of work studies auction-design in MAB formalisms, often motivated by applications in ad auctions (Babaioff et al., 2009; Devanur and Kakade, 2009; Babaioff et al., 2015). In these models, in every round the auctioneer selects one advertiser's item, which is subsequently clicked or not, and the goal of the auctioneer is to incentivize advertisers to truthfully bid their value-per-click by constructing selection and payment rules.

To the best of our knowledge, our work is the first to study the situation where the arms' strategies (as well as other parameters) are initially unobserved, and must be learned from interaction while simultaneously incentivizing arms under uncertainty without payments. As a result, while other work is usually able to precisely incentivize certain arm strategies, our mechanism design and characterization of the Nash equilibria are *approximate*.

## 3 THE STRATEGIC CLICK-BANDIT PROBLEM

We consider a natural strategic variant of the classical MAB, motivated by applications in online recommendation. Unlike classical MABs, strategic click-bandits feature decentralized interactions with the learner and multiple self-interested arms.

Let $[K] := \{1, \ldots, K\}$ denote the set of arms, each being viewed as a strategic *agent*. The strategic click-bandit proceeds in two phases. In the first phase, the learner commits to an online learning policy $M$, upon which each arm $i$ chooses a description, which results in a corresponding click-rate $s_i \in [0, 1]$. The second phase proceeds in rounds. At each round $t$: (1) the algorithm $M$ pulls/recommends an arm $i_t$ based on observed past data; (2) arm $i_t$ is clicked with probability $s_{i_t}$; (3) if $i_t$ is clicked, arm $i_t$ receives utility 1 (whereas all other arms $i$ receive utility 0) and the learner observes a post-click reward $r_{t,i_t} \in [0, 1]$ drawn from $i_t$'s reward distribution with mean $\mu_{i_t} \in [0, 1]$. If $i_t$ is *not* clicked, all arms receive 0 utility and the learner does not observe any post-click rewards. The post-click mean $\mu_i$ is fixed for each arm $i$ and captures the *true value* of the arm. From the learner's perspective, *both* $s_i$ and $\mu_i$ of each arm are unknown but can be learned from online bandit feedback, that is, whether the recommended arm is clicked and, if so, what its realized reward is. In the following, we will also refer to the online learning policy $M$ as a *mechanism* to emphasize its dual role in learning and incentive design. We summarize the interaction in Model 1.

### 3.1 LEARNER'S UTILITY

The learner's utility of selecting an arm $i$ with CTR $s_i$ and post-click value $\mu_i$ is denoted $u(s_i, \mu_i)$. One example of this utility function is $u(s, \mu) = s\mu$. In this case, the learner monotonically prefers large $s$ and does not care about how much the click-rate $s$ differs from the post-click value $\mu$. However, we believe that the learner (e.g., a platform like Youtube or Airbnb) usually values consistency between the click-rates and the post-click values of arms. This could be captured by a penalty term

for how much $s_i$ differs from $\mu_i$; for instance, a natural choice is $u(s, \mu) = s\mu - \lambda(s - \mu)^2$ for some weight $\lambda > 0$. Such *non-monotonicity* of the learner's utility $u(s_i, \mu_i)$ in $s_i$ *versus* arm $i$'s monotonic preference of larger click-rates forms the fundamental tension in the strategic click-bandit model and is also the reason that mechanism design is needed. We keep the above utility functions in mind as running examples, but derive our results for a much more general class of functions satisfying the following mild regularity assumptions:

(A1) $u \colon [0, 1] \times [0, 1] \to \mathbb{R}$ is $L$-Lipschitz w.r.t. the $\ell_1$-norm.

(A2) $u^*(\mu) := \max_{s \in [0, 1]} u(s, \mu)$ is monotonically increasing.

(A3) $s^*(\mu) := \mathrm{argmax}_{s \in [0, 1]} u(s, \mu)$ is $H$-Lipschitz and is bounded away from zero.

Assumption (A1) bounds the loss of selecting a suboptimal arm. (A2) states that, in the (idealized) situation when the arms choose click-rates so as to maximize the learner's utility $u$, then arms with larger post-click rewards $\mu$ are always preferred. (A3) then ensures that from the perspective of the learner most desired strategy $s^*(\mu)$ does not change abruptly w.r.t. $\mu$ and the learner wishes to incentivize non-zero click-rates. In what follows, the function $s^*(\mu)$ will play a central role as it describes the arm strategy that maximizes the learner's utility. For instance, in the case of $u(s, \mu) = s\mu - \lambda(s - \mu)^2$ it is given by $s^*(\mu) = (1 + \frac{1}{2\lambda})\mu$. As such, the learner will typically try to incentivize an arm with post-click reward $\mu_i$ to choose strategy $s^*(\mu_i)$.

## 3.2 ARMS' UTILITY AND NASH EQUILIBRIA AMONG ARMS

The mean post-click reward $\mu_i$ of each arm $i$ is fixed, whereas arm $i$ can freely choose the CTR $s_i$. In the strategic click-bandit, the objective of each arm $i$ is to maximize the number of times it gets clicked $\sum_{t=1}^{T} \mathbb{1}_{\{i_t = i\}} c_{t,i}$, which captures the objectives of vendors on internet platforms for whom user traffic typically proportionally converts to revenue.[3] We now introduce the solution concept for the game among arms defined by a mechanism $M$ and post-click rewards $\mu_1, \ldots, \mu_K$, often referred to as an *equilibrium*. Let $s_{-i}$ denote the $K - 1$ strategies of all arms except $i$. Each arm $i$ chooses $s_i$ to maximize their *expected* number of clicks $v_i(M, s_i, s_{-i})$, which is a function of the mechanism $M$, their own action $s_i$ as well as all other arms' actions $s_{-i}$. Concretely,

$$v_i(M, s_i, s_{-i}) := \mathbb{E}_M \left[ \sum_{t=1}^{T} \mathbb{1}_{\{i_t = i\}} c_{t,i} \right] \tag{1}$$

where the expectation is taken over the mechanism's decisions and the environment's randomness. We generally write $s := (s_1, \ldots, s_K)$ to summarize a strategy profile of the arms. Let $\Sigma$ denote the set of probability measures over $[0, 1]$. Given a *mixed* strategy profile $\boldsymbol{\sigma} = (\sigma_i, \sigma_{-i}) \in \Sigma^K$, i.e., a distribution over $[0, 1]^K$, arm $i$'s utility is then defined as $v_i(M, \sigma_i, \sigma_{-i}) := \mathbb{E}_{s \sim \boldsymbol{\sigma}}[v_i(M, s_i, s_{-i})]$.

**Definition 3.1** (Nash Equilibrium). We say that $\boldsymbol{\sigma} = (\sigma_1, \ldots, \sigma_K) \in \Sigma^K$ is a Nash equilibrium (NE) under mechanism $M$ if $v_i(M, \sigma_i, \sigma_{-i}) \geq v_i(M, \sigma_i', \sigma_{-i})$ for all $i \in [K]$ and strategies $\sigma_i' \in \Sigma$.

In other words, $\boldsymbol{\sigma}$ is in NE if no arm can increase its utility by *unilaterally* deviating to some other strategy. If some NE $\boldsymbol{\sigma} \in \Sigma^K$ has weight one on a pure strategy profile $s \in [0, 1]^K$, this equilibrium is said to be in pure-strategies. Let $\mathrm{NE}(M) := \{ \boldsymbol{\sigma} \in \Sigma^K : \boldsymbol{\sigma} \text{ is a NE under } M \}$ denote the set of all (possibly mixed) NE under mechanism $M$. Following conventions in standard economic analysis, we assume that the arms will form a NE in $\mathrm{NE}(M)$ in response to an algorithm $M$.[4]

**Remark 3.1** (Existence of Nash Equilibrium). *In general, the arms' utility functions $v_i(M, s_i, s_{-i})$ may be discontinuous in the arms' strategies due to their intricate dependence on the learning algorithm $M$. It is well-known that in games with discontinuous utilities, a NE may not exist (Reny, 1999). However, for all subsequently considered algorithms we will prove the existence of a NE by either explicitly describing the equilibrium or implicitly proving its existence.*

---

[3]More generally, different arms $i$ may have a different value-per-click $\nu_i$ that could as well depend on $\mu_i$ so that $v_i(M, s_i, s_{-i}) = \mathbb{E}_M[\sum_t \mathbb{1}_{\{i_t = i\}} c_{t,i} \nu_i]$. This can easily be accommodated for by our model and our results readily extend to this case since each arm's goal still boils down to maximizing the number of clicks.

[4]For instance, a sufficient condition for the arms to find a NE is their knowledge about how far away they are from the best arm, i.e., their optimality gap in post-click rewards $\Delta_i := \max_{j \in [K]} \mu_j - \mu_i$.

### 3.3 STRATEGIC REGRET

The learner's goal is to maximize $\sum_{t=1}^{T} u(s_{i_t}, \mu_{i_t})$ which naturally depends on the arm strategies $s_1, \ldots, s_K$. For given post-click values $\mu_1, \ldots, \mu_K$, the maximal utility $u(s^*, \mu^*)$ is then achieved for $\mu^* := \max_{i \in [K]} \mu_i$ and $s^* := s^*(\mu^*)$, that is, $u(s^*, \mu^*) = \max_{i \in [K]} \max_{s \in [0,1]} u(s, \mu_i)$. With $u(s^*, \mu^*)$ as a benchmark, we can define the *strategic regret* of a mechanism $M$ under a pure-strategy equilibrium $\boldsymbol{s} \in \text{NE}(M)$ as

$$R_T(M, \boldsymbol{s}) := \mathbb{E}\left[\sum_{t=1}^{T} u(s^*, \mu^*) - u(s_{i_t}, \mu_{i_t})\right]. \tag{2}$$

For some mixed-strategy equilibrium $\boldsymbol{\sigma} \in \text{NE}(M)$, we then accordingly define strategic regret as $R_T(M, \boldsymbol{\sigma}) := \mathbb{E}_{\boldsymbol{s} \sim \boldsymbol{\sigma}}[R_T(M, \boldsymbol{s})]$. In general, there may exist several Nash equilibria for the arms under a given mechanism $M$. We can then consider the *strong strategic regret* of $M$ given by the regret under the worst-case equilibrium:

$$R_T^+(M) := \max_{\boldsymbol{\sigma} \in \text{NE}(M)} R_T(M, \boldsymbol{\sigma}),$$

or the *weak strategic regret* given by the regret under the most favorable equilibrium:

$$R_T^-(M) := \min_{\boldsymbol{\sigma} \in \text{NE}(M)} R_T(M, \boldsymbol{\sigma}),$$

where $R_T^-(M) \leq R_T^+(M)$. The regret upper bound of our proposed algorithm, UCB-S, holds under any equilibrium in $\text{NE}(\text{UCB-S})$, thereby bounding *strong strategic regret* (Theorem 5.3). On the other hand, the proven lower bounds (Proposition 4.1 and Theorem 5.5) hold for *weak strategic regret* and thus also apply to its strong counterpart.

## 4 LIMITATIONS OF INCENTIVE-UNAWARE ALGORITHMS

We start our analysis of the strategic click-bandit problem by showing that simply finding the arm with the largest post-click reward, $\text{argmax}_i \mu_i$, or largest utility, $\text{argmax}_i u(s_i, \mu_i)$, is insufficient to achieve $o(T)$ *weak* strategic regret. In fact, we find that even with oracle knowledge of $\mu_1, \ldots, \mu_K$ and $s_1, \ldots, s_K$, an algorithm may suffer linear weak strategic regret if it fails to account for the arms' strategic nature. For such incentive-*unaware* oracle algorithms, we show a $\Omega(T)$ lower bound for weak strategic regret on any non-trivial problem instance.

Recall that $\mu^* := \max_{i \in [K]} \mu_i$ and $s^* := s^*(\mu^*)$ and suppose that the arm $i^* = \text{argmax}_{i \in [K]} \mu_i$ with maximal post-click rewards is unique. Our negative results rely on the following problem-dependent gaps in terms of utility:

$$\beta := u(s^*, \mu^*) - u(1, \mu^*) \quad \text{and} \quad \eta := u(s^*, \mu^*) - \max_{i \in [K] \setminus \{i^*\}} u^*(\mu_i).$$

Here, $\beta$ denotes the cost of the optimal arm $i^*$ deviating from the desired strategy $s^* = s^*(\mu^*)$ by playing $s_{i^*} = 1$. The quantity $\eta$ denotes the gap between the maximally achievable utility $u(s^*, \mu^*)$ and the utility of the second best arm.

**Proposition 4.1.** *Let $\mu$-Oracle be the algorithm with oracle knowledge of $\mu_1, \ldots, \mu_K$ that plays $i_t = \text{argmax}_{i \in [K]} \mu_i$ in every round $t$, whereas $(s, \mu)$-Oracle is the algorithm with oracle knowledge of $\mu_1, \ldots, \mu_K$ and $s_1, \ldots, s_K$ that always plays $i_t = \text{argmax}_{i \in [K]} u(s_i, \mu_i)$ with ties broken in favor of the larger $\mu$. We then have*

(i) *Under every equilibrium $\boldsymbol{\sigma} \in \text{NE}(\mu$-Oracle$)$, the $\mu$-Oracle suffers regret $\Omega(\beta T)$, i.e.,*

$$R_T^-(\mu\text{-Oracle}) = \Omega(\beta T).$$

(ii) *Under every $\boldsymbol{\sigma} \in \text{NE}((s, \mu)$-Oracle$)$, the $(s, \mu)$-Oracle suffers regret $\Omega(\min\{\beta, \eta\}T)$, i.e.,*

$$R_T^-((s, \mu)\text{-Oracle}) = \Omega(\min\{\beta, \eta\}T).$$

---

**Mechanism 1:** UCB with Screening (UCB-S)

---

1 **initialize:** $A_0 = [K]$

2 **for** $t = 1, \ldots, T$ **do**

3    **if** $A_{t-1} \neq \emptyset$ **then**

4      Select $i_t \in \mathrm{argmax}_{i \in A_{t-1}} \overline{\mu}_i^{t-1}$

5    **else**

6      Select $i_t$ uniformly at random from $[K]$

7    Arm $i_t$ is clicked with probability $s_{i_t}$, i.e., $c_{t,i_t} \sim \mathrm{Bern}(s_{i_t})$

8    **if** $i_t$ was clicked ($c_{t,i_t} = 1$) **then**

9      Observe post-click reward $r_{t,i_t}$

10    **if** $\overline{s}_{i_t}^t < \min_{\mu \in [\underline{\mu}_{i_t}^t, \overline{\mu}_{i_t}^t]} s^*(\mu)$ or $\underline{s}_{i_t}^t > \max_{\mu \in [\underline{\mu}_{i_t}^t, \overline{\mu}_{i_t}^t]} s^*(\mu)$ **then**

11      Ignore arm $i_t$ in future rounds: $A_t \leftarrow A_{t-1} \setminus \{i_t\}$

---

*Proof Sketch. (i)*: We show that $s = 1$ is a strictly dominant strategy for arm $i^*$ under the $\mu$-Oracle. This implies that arm $i^*$ plays $s_{i^*} = 1$ with probability one in every NE under the $\mu$-Oracle. The claimed lower bound then follows from bounding the instantaneous regret per round from below by $\beta$. *(ii)*: Let $j^* \in \mathrm{argmax}_{i \neq i^*} \mu_i$. It can be seen that in any NE, arm $i^*$ will play the largest $s \in [0, 1]$ such that $u(s, \mu_{i^*}) \geq u(s_{j^*}, \mu_{j^*})$. We then show that either $s_{i^*} = 1$ or $u(s_{i^*}, \mu_{i^*}) = u(s^*(\mu_{j^*}), \mu_{j^*})$. Once again this allows us to lower bound the regret per round by $\min\{\beta, \eta\}$. □

As a concrete example of the failure of the $\mu$-Oracle and the $(s, \mu)$-Oracle, let us consider the running example of $u(s, \mu) = s\mu - \lambda(s - \mu)^2$. In this case, letting $\lambda = 5$ and $\mu_{i^*} = 0.8$ and $\mu_i \leq 0.7$ for $i \neq i^*$, we get $\beta \geq 0.1$ and $\eta \geq 0.1$ so that both oracles suffer $\Omega(T)$ regret in every equilibrium.

## 5 No-Regret Incentive-Aware Learning: UCB-S

The results of Proposition 4.1 suggest that any incentive-unaware learning algorithm that is oblivious to the strategic nature of the arms will generally fail to achieve low regret. In particular, "unconditional" selection of any arm will likely result in undesirable equilibria among arms. For these reasons, we deploy a conceptually simple screening idea, which threatens arms with elimination when deviating from the desired strategies.

Let denote $n_t(i)$ be the number of times up to (and including) round $t$ that arm $i$ was selected by the learner, and let $m_t(i)$ denote the number of times post-click rewards were observed for arm $i$ up to (and including) round $t$. Let $\widehat{s}_i^t$ be the average observed click-rate and $\widehat{\mu}_i^t$ the average observed post-click reward for arm $i$. We then define the pessimistic and optimistic estimates of $s_i$ and $\mu_i$ as

$$\underline{s}_i^t = \widehat{s}_i^t - \sqrt{2\log(T)/n_t(i)}, \qquad \overline{s}_i^t = \widehat{s}_i^t + \sqrt{2\log(T)/n_t(i)},$$
$$\underline{\mu}_i^t = \widehat{\mu}_i^t - \sqrt{2\log(T)/m_t(i)}, \qquad \overline{\mu}_i^t = \widehat{\mu}_i^t + \sqrt{2\log(T)/m_t(i)}.$$

where $\underline{s}_i^t = -\infty$ and $\overline{s}_i^t = +\infty$ for $n_t(i) = 0$ as well as $\underline{\mu}_i^t = -\infty$ and $\overline{\mu}_i^t = +\infty$ for $m_t(i) = 0$.

In every round, UCB-S (Mechanism 1) selects arms optimistically according to their post-click rewards and subsequently observes if the arm is clicked, i.e., $c_{t,i_t}$, and, if so, a post-click reward $r_{t,i_t}$. However, if an arm's click-rate $s_i$ is detected to be different from the learner's desired arm strategy $s^*(\mu_i)$, the arm is eliminated forever, expressed by the screening rule in line 10:

$$\overline{s}_{i_t}^t < \min_{\mu \in [\underline{\mu}_{i_t}^t, \overline{\mu}_{i_t}^t]} s^*(\mu) \quad \text{or} \quad \underline{s}_{i_t}^t > \max_{\mu \in [\underline{\mu}_{i_t}^t, \overline{\mu}_{i_t}^t]} s^*(\mu).$$

The only exception is when all arms have been eliminated. Then, UCB-S plays them all uniformly for the remaining rounds. To ensure that the elimination of an arm is credible and justified with high probability, we leverage confidence bounds on $s_i$ and $\mu_i$. More precisely, if an arm is truthful and chooses $s_i = s^*(\mu_i)$, then with probability $1 - 1/T^2$ it will not be eliminated by the screening rule.

As a prelude to the analysis of the UCB-S mechanism, we begin by showing that there always exists a NE among the arms under UCB-S. As mentioned briefly in Section 3, the existence of a NE among the arms is not guaranteed under an arbitrary mechanism due to the arms' continuous strategy space and possibly discontinuous utility function.

**Lemma 5.1.** *For any post-click rewards $\mu_1, \ldots, \mu_K$, there always exists a (possibly mixed) Nash equilibrium for the arms under the UCB-S mechanism.*

### 5.1 CHARACTERIZING THE NASH EQUILIBRIA UNDER UCB-S

We now approximately characterize all NE for the arms under the UCB-S mechanism. In order to prove a regret upper bound for UCB-S, it will be key to ensure that each arm $i$ plays a strategy $s_i$ which is sufficiently close to the desired strategy $s^*(\mu_i)$ (i.e., the strategy that maximizes the learner's utility). This is particularly important for arms $i^*$ with maximal post-click rewards $\mu_{i^*} = \max_{i \in [K]} \mu_i$. If such arms $i^*$ were to deviate substantially from $s^*(\mu_{i^*})$, e.g., by a constant amount, the learner would be forced to suffer constant regret even when selecting arms with maximal post-click rewards, making it impossible to achieve sublinear regret.

In the following, we show that under the UCB-S mechanism every NE is such that the strategies of arms with maximal post-click rewards deviate from the desired strategies by at most $\widetilde{\mathcal{O}}(\sqrt{K/T})$. We then also show that for suboptimal arms the difference between each arm $i$'s strategy $s_i$ and the desired strategy $s^*(\mu_i)$ is governed by their optimality gap in post-click rewards, given by $\Delta_i := \mu^* - \mu_i$. Recall that $H$ denotes the Lipschitz constant of $s^*(\mu)$.

**Theorem 5.2.** *For all $s \in \mathrm{supp}(\boldsymbol{\sigma})$ with $\boldsymbol{\sigma} \in \mathrm{NE}(\mathrm{UCB\text{-}S})$ and all $i \in [K]$:*

$$s_i = s^*(\mu_i) + \mathcal{O}\left( H \cdot \max\left\{ \Delta_i, \sqrt{\frac{K \log(T)}{T}} \right\} \right).$$

*In particular, for all arms $i^* \in [K]$ with $\Delta_{i^*} = 0$, i.e., maximal post-click rewards:*

$$s_{i^*} = s^*(\mu_{i^*}) + \mathcal{O}\left( H \sqrt{\frac{K \log(T)}{T}} \right).$$

The derivation of Theorem 5.2 can be best understood by noting that the estimates of each arm's strategy roughly concentrate at a rate of $1/\sqrt{t}$. Then, depending on how often an arm expects to be selected by UCB-S, it can exploit our uncertainty about its strategy and safely increase its click-rates to match our confidence. Generally, optimal arms expect at least $T/K$ allocations while preventing elimination, which can be seen to imply NE strategies that deviate by at most $\sqrt{K/T}$. On the other hand, suboptimal arms can expect roughly $\log(T)/\Delta_i^2$ allocations as long as they can prevent elimination and all other arms act rationally, which results in the linear dependence on $\Delta_i$. Hence, interestingly UCB-S' selection policy directly impacts the truthfulness of the arms, as arms that are selected more frequently are forced to choose strategies closer to $s^*(\mu_i)$. We thus observe a trade-off between incentivizing *all* arms to be truthful and recommending only the best arms. The proof of Theorem 5.2 (Appendix C) then relies on the above observation and careful and repeated application of the best response property of the Nash equilibrium.

### 5.2 UPPER BOUND OF THE STRONG STRATEGIC REGRET OF UCB-S

With the approximate NE characterization from Theorem 5.2 at our disposal, we are ready to prove a regret upper bound for UCB-S. We show that the *strong strategic regret* of the UCB-S mechanism is upper bounded by $\widetilde{\mathcal{O}}(\sqrt{KT})$, that is, for any $\boldsymbol{\sigma} \in \mathrm{NE}(\mathrm{UCB\text{-}S})$ the regret guarantee holds.

**Theorem 5.3.** *Let $\Delta_i := \mu^* - \mu_i$ and let $L$ and $H$ denote the Lipschitz constants of $u(s, \mu)$ and $s^*(\mu)$, respectively. The strong strategic regret of UCB-S is bounded as*

$$R_T^+(\mathrm{UCB\text{-}S}) = LH \cdot \mathcal{O}\left( \sqrt{KT \log(T)} + \sum_{i : \Delta_i > 0} \frac{\log(T)}{\Delta_i} \right). \tag{3}$$

*In other words, the above regret bound is achieved under any equilibrium $\boldsymbol{\sigma} \in \mathrm{NE}(\mathrm{UCB\text{-}S})$.*

*Proof Sketch.* As suggested by the regret bound there are two sources of regret. Broadly speaking, the first term on the right hand side of (3) corresponds to the regret UCB-S suffers due to arms with maximal post-click rewards (i.e., $\Delta_i = 0$) deviating from the utility-maximizing strategy $s^*(\mu^*)$. For such arms Theorem 5.2 bounded the deviation by a term of order $\sqrt{K/T}$, thereby leading to at most order $\sqrt{KT}$ regret. The second term in (3) corresponds to the regret suffered from playing arms with suboptimal post-click rewards, i.e., $\Delta_i > 0$. Using a typical UCB argument, the Lipschitzness of $u(s, \mu)$ and $s^*(\mu)$, and again Theorem 5.2 applied to $|s^*(\mu^*) - s_i| \leq |s^*(\mu^*) - s^*(\mu_i)| + \mathcal{O}(H\Delta_i) \leq H\Delta_i + \mathcal{O}(H\Delta_i)$ we obtain the claimed upper bound. □

Similarly to classical MABs we can state a regret bound independent of the instance-dependent quantities $\Delta_i$ and translate Theorem 5.3 into a minimax-type guarantee.

**Corollary 5.4.** *The strong strategic regret of UCB-S is bounded as*

$$R_T^+(\text{UCB-S}) = \mathcal{O}\left(LH\sqrt{KT\log(T)}\right).$$

*In other words, the above regret bound is achieved under any equilibrium $\boldsymbol{\sigma} \in \text{NE}(\text{UCB-S})$.*

Theorem 5.3 nicely shows that the additional cost of the incentive design and the strategic behavior of the arms is of order $\sqrt{KT}$ which primarily stems from arms with maximal post-click rewards deviating by roughly $\sqrt{K/T}$ from the desired strategy (see Theorem 5.2). The dishonesty of suboptimal arms does not notably contribute to the regret and is contained in the $\log(T)/\Delta_i$ expressions as we can bound the number of times suboptimal arms are played sufficiently well. As a result, the total cost of incentive design and strategic behavior matches the minimax learning complexity of MABs so that we obtain an overall $\tilde{\mathcal{O}}(\sqrt{KT})$ strategic regret bound under every equilibrium.

### 5.3 Lower Bound for Weak Strategic Regret

Complementing our regret analysis, we prove a lower bound on *weak strategic regret* in the strategic click-bandit. By definition, weak strategic regret lower bounds its strong counterpart, i.e., $R_T^-(M) \leq R_T^+(M)$, so that the shown lower bound directly applies to strong strategic regret as well, which implies that UCB-S is near-optimal.

**Theorem 5.5.** *Let $M$ be any mechanism with $\text{NE}(M) \neq \emptyset$. There exists a utility function $u$ satisfying (A1)-(A3) and post-click rewards $\mu_1, \ldots, \mu_K$ such that for all Nash equilibria $\boldsymbol{\sigma} \in \text{NE}(M)$:*

$$R_T(M, \boldsymbol{\sigma}) = \Omega\big(\sqrt{KT}\big).$$

*In other words, $R_T^-(M) = \Omega\big(\sqrt{KT}\big)$.*

*Proof Sketch.* Consider the utility function $u(s, \mu) = s\mu$. Intuitively, for any low regret mechanism $M$ the NE for the arms will be in $(s_1, \ldots, s_K) = (1, \ldots, 1)$ as these strategies maximize the learner's utility $u$ and are to the advantage of the arms. In this case, the learning problem reduces to a classical MAB and we inherit the well-known minimax $\sqrt{KT}$ lower bound. However, it is not directly clear that there exists no better mechanism that would, e.g., incentivize arm strategies $(s_1, \ldots, s_{i^*}, \ldots, s_K) = (0, \ldots, 1, \ldots, 0)$ under which $i^* = \arg\max_i \mu_i$ becomes easier to distinguish from $i \neq i^*$. For this reason, we argue via the arms' utilities and lower bound the minimal utility a suboptimal arm must receive in any NE. This directly implies a lower bound on the number of times we must play any suboptimal arm in equilibrium, which yields the claimed result. □

## 6 Simulating Strategic Arm Behavior via Repeated Interaction

Goal of the experiments is to analyze the effect of the proposed incentive-aware learning algorithm UCB-S on strategically responding arms. Strategic arm behavior is here modeled through decentralized gradient ascent and repeated interaction with the mechanism. Contrary to the assumption of arms playing in NE, arms follow a simple gradient ascent strategy to adapt to the mechanism, which serves as a realistic and natural model of strategic behavior. This requires no prior knowledge from the point of view of the arms and all learning is performed through sequential interaction with the mechanism. For this reason, the final strategies in our experiments may not necessarily be in NE. Despite this, we want to see whether the mechanism is still able to incentivize arms to behave in the desired manner which will also provide insight into the robustness of the proposed incentive design.

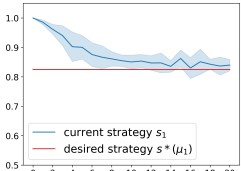 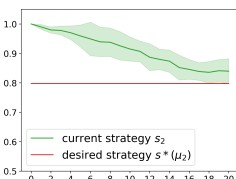 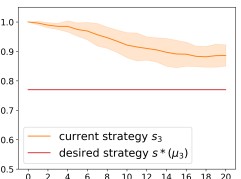 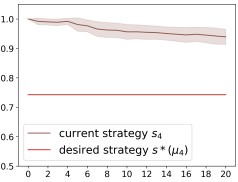

(a) Optimal arm with mean $\mu_1 = 0.75$.

(b) Suboptimal arm with mean $\mu_2 = 0.725$.

(c) Suboptimal arm with mean $\mu_3 = 0.7$.

(d) Suboptimal arm with mean $\mu_4 = 0.675$.

Figure 2: The strategic behavior of $K = 4$ arms when each arm uses gradient ascent to maximize their utility $v_i$ in response to the UCB-S mechanism. In red, the desired strategy $s^*(\mu_i)$ for each arm $i$, respectively. As suggested by Theorem 5.2, the truthfulness, i.e., distance to $s^*(\mu_i)$, of a suboptimal arm $i$ is governed by the arm's optimality gap $\Delta_i$. We see this confirmed as the distance $s_i - s^*(\mu_i)$ increases as $\Delta_i$ increases. In accordance with our theoretical results, the optimal arm $1$ has the largest incentive to play close to the desired strategy (as it loses the most when eliminated).

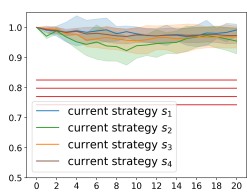 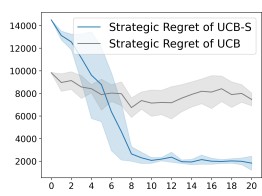

Figure 3: Strategic arm behavior when interacting with incentive-*unaware* standard UCB.

Figure 4: Strategic regret of UCB-S and standard UCB as arms adapt their strategies.

**Experimental Setup.** We consider the earlier introduced utility function defined as $u(s, \mu) = s\mu - \lambda(s - \mu)^2$ such that the desired (learner's utility-maximizing) strategy given $\mu$ is $s^*(\mu) = (1 + \frac{1}{2\lambda})\mu$. We let $\lambda = 5$. To model the strategic behavior of arms in response to UCB-S, we let the strategic arms interact with the mechanism over the course of 20 epochs (x-axis) and model each arm's strategic behavior via gradient ascent w.r.t. its utility $v_i$. More precisely, after every epoch (i.e., interaction over $T = 50k$ rounds), each arm performs an approximated gradient step with respect to its utility $v_i$. We initialized the arm strategies to $s_i = 1$, however, our experiments show that other initialization, such as $s_i = 0$ or $s_i = 0.5$, yield similar results. All results are averaged over 10 complete runs and the standard deviation shown in shaded color.

**Results.** The conducted simulations show that under natural greedy behavior as modeled by gradient ascent, the incentive design of UCB-S is still effective and desirable arm strategies incentivized (Figure 2). Most notably, the optimal arm (having the largest incentive to be truthful) converges to a strategy close to the desired strategy $s^*(\mu_1)$. The suboptimal arms do not converge to a strategy close to the desired strategy and we observe that the distance to $s^*(\mu_i)$ depends on the optimality gap $\Delta_i$, which mirrors our theoretical results (Theorem 5.2). In addition, Figure 4 shows that as the arms interact with UCB-S and adapt their strategies, the regret of UCB-S improves substantially. In contrast, incentive-unaware algorithms like UCB fail to incentivize desirable strategies (all arm strategies remain close to 1, see Figure 3) and UCB accordingly suffers large regret (Figure 4) throughout all epochs. The observation that UCB-S initially suffer larger regret than UCB can be explained by the elimination rule causing UCB-S to select arms uniformly at random when arms are notably untruthful. This threat of elimination, however, incentivizes the arms to adapt their strategies in the next epoch and eventually leads to smaller regret for UCB-S.

# 7 DISCUSSION

We study the strategic click-bandit problem in which each arm is associated with a click-rate, chosen strategically by the arms, and an immutable post-click reward. We show the necessity of incentive design in this model and design an incentive-aware online learning algorithm that incentivizes desirable arm strategies under uncertainty. As the learner has no prior knowledge of the arm strategies and the post-click rewards, the mechanism design is approximate and leaves room for arms to exploit the learner's uncertainty. This leads to an interesting regret bound which makes the intuition precise that arms can exploit the learner's uncertainty about their strategies. In our simulations we then observe that our incentive design is robust and still effective under natural greedy arm behavior and that the design of incentive-aware learning algorithms is necessary to achieve low regret under strategic arm behavior. Some interesting open questions which we leave for future work include whether the proposed incentive design remains effective under adaptive arm strategies and whether we can construct a mechanism under which there exists a desirable NE in dominant strategies.

**Acknowledgments.** We thank the anonymous reviewers for their insightful and constructive comments. We also want to thank Boi Faltings for helpful discussions in the early stages of this work. Thomas Kleine Buening was partly supported by the Norwegian Research Council (Grant 302203). Haifeng Xu is supported by the AI2050 program at Schmidt Sciences (Grant G-24-66104), an NSF Award CCF-2303372, an Army Research Office Award W911NF-23-1-0030, and an Office of Naval Research Award N00014-23-1-2802.

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

APPENDIX

The appendix is arranged as follows:

## A  PROOF OF PROPOSITION 4.1

***Proof of Proposition 4.1.*** ***(i)***: Under any strategy profile $s = (s_1, \ldots, s_K)$, arm $i \neq i^*$ has utility $v_i(\mu\text{-Oracle}, s_i, s_{-i}) = 0$, while arm $i^*$ has utility

$$v_{i^*}(\mu\text{-Oracle}, s_{i^*}, s_{-i^*}) = T s_{i^*}.$$

Hence, the pure strategy $s = 1$ is a strictly dominant strategy for arm $i^*$, which implies that $i^*$ plays $s_{i^*} = 1$ with probability one in every Nash equilibrium. Now,

$$u(s^*, \mu_{i^*}) - u(s_{i^*}, \mu_{i^*}) = u(s^*, \mu_{i^*}) - u(1, \mu_{i^*}) = \beta$$

and the $\mu$-Oracle thus suffers regret $\beta$ every round, which implies the claimed $\Omega(\beta T)$ lower bound in every equilibrium.

***(ii)***: Let $j^* \in \arg\max_{i \neq i^*} \mu_i$ be the arm with second largest post-click value and define $u_{j^*}^* := u(s^*(\mu_{j^*}), \mu_{j^*})$. Let $s'$ be the largest $s \in [0, 1]$ such that $u(s', \mu_{i^*}) \geq u_{j^*}^*$. We distinguish between two cases:

**Case 1.** Suppose that $u(s', \mu_{i^*}) > u_{j^*}^*$. From the continuity of $u$ it then follows that $s' = 1$. To see this, suppose the contrary is true. Then, for all $s'' > s'$ with $s'' \in [0, 1]$ it must hold that $u(s'', \mu_{i^*}) < u_{j^*}^*$ by definition of $s'$ as the largest $s \in [0, 1]$ such that $u(s, \mu_{i^*}) \geq u_{j^*}$. However, this contradicts the continuity of $u(s, \mu)$ in $s$, since we have just shown that $u(s'', \mu_{i^*}) < u_{j^*}^* < u(s', \mu_{i^*})$ for all $s'' > s'$. We have thus shown by contradiction that $s' = 1$.

Then, if arm $i^*$ chooses strategy $s_{i^*} = 1$, arm $i^*$ is pulled every round by $(s, \mu)$-Oracle for all $s_{-i^*} \in [0, 1]^{K-1}$ so that $v_{i^*}((s, \mu)\text{-Oracle}, 1, s_{-i^*}) = T$. This immediately implies that $s_{i^*} = 1$ is a strictly dominant strategy for $i^*$, since $v_{i^*}((s, \mu)\text{-Oracle}, s, s_{-i^*}) \leq T s < T$ for all $s \in [0, 1)$. Thus, arm $i^*$ plays $s_{i^*} = 1$ in every Nash equilibrium of the arms. Analogous to the proof of (i), this yields $|u(s^*, \mu^*) - u(s_{i^*}, \mu_{i^*})| = \beta$, which implies that the $(s, \mu)$-Oracle suffers $\Omega(\beta T)$ under any Nash equilibrium of the arms.

**Case 2.** Suppose that $u(s', \mu_{i^*}) = u_{j^*}^*$. In a first step, we show that arm $i^*$ plays $s'$ with probability one in every Nash equilibrium. We begin by noting that if arm $i^*$ plays $s_{i^*} = s'$, then for any opponent strategies $s_{-i^*} \in [0, 1]^{K-1}$ arm $i^*$ is played all $T$ rounds so that $v_{i^*}((s, \mu)\text{-Oracle}, s', s_{-i^*}) = T s'$. Naturally, $s'$ thus strictly dominates any other strategy $s'' < s'$, since $v_i((s, \mu)\text{-Oracle}, s'', s_{i^*}) \leq T s''$.

Next, suppose that arm $i^*$ plays some strategy $s'' > s'$ with probability one.[5] Then, by definition of $s'$, we have $u(s'', \mu_{i^*}) < u_{j^*}^* := u(s^*(\mu_{j^*}), \mu_{j^*})$. As a result, arm $j^*$'s best response $s_{j^*}$ to $s''$

---

[5] For simplicity, we assume that arm $i^*$ plays the strategy with probability one. The case where $i^*$ plays $s'' > s'$ with some positive probability can be treated analogously.

will be such that $u(s'', \mu_{i^*}) < u(s_{j^*}, \mu_{j^*})$, thereby obtaining utility $v_{j^*}((s, \mu)\text{-Oracle}, s_{j^*}, s_{-j^*}) \geq Ts^*(\mu_{j^*})$. As a result, if $j^*$ plays a best response, arm $i^*$ receives utility $0$ when playing $s''$, whereas arm $i^*$ receives utility $Ts'$ when playing $s'$. Hence, any $s'' > s'$ cannot be part of an equilibrium for arm $i^*$ and we have shown that arm $i^*$ plays $s'$ with probability one in every equilibrium. Finally, by definition of $s'$, we have

$$u(s^*, \mu^*) - u(s', \mu_{i^*}) \geq u(s^*, \mu^*) - u(s^*(\mu_{j^*}), \mu_{j^*}) = u(s^*, \mu^*) - u^*(\mu_{j^*}) = \eta$$

which implies that $(s, \mu)$-Oracle suffers $\Omega(\eta T)$ regret under any Nash equilibrium of the arms. Hence, we obtain the claimed lower bound of $\Omega\big(\min\{\beta, \eta\}T\big)$.

$\square$

**Remark A.1.** *Interestingly, when the $(s, \mu)$-Oracle from Proposition 4.1 (ii) does not break ties in favor of the larger $\mu$ but instead uniformly at random, it can be shown that in all but a few problem instances no Nash equilibrium for the arms exists. However, for any $\varepsilon > 0$ we can explicitly construct an $\varepsilon$-Nash equilibrium for the arms under which the algorithm suffers $\Omega(\min\{\beta, \eta\ T)$ strategic regret.*

Before proving the statement of Remark A.1, we formally introduce the concept of an $\varepsilon$-Nash equilibrium among the arms here.

**Definition A.1** ($\varepsilon$-Nash Equilibrium). For $\varepsilon > 0$, we say that strategies $\boldsymbol{\sigma} = (\sigma_1, \ldots, \sigma_K)$ form an $\varepsilon$-Nash equilibrium under $M$ if $v_i(M, \sigma_i, \sigma_{-i}) \geq v_i(M, \sigma_i', \sigma_{-i}) - \varepsilon$ for all $i \in [K]$ and $\sigma_i' \in \Sigma$.

For Remark A.1, we will show that there exists an $\varepsilon$-Nash equilibrium in pure-strategies $\boldsymbol{s} \in [0, 1]^K$ such that the oracle algorithm that breaks ties uniformly suffers linear strategic regret.

***Proof of Remark A.1.*** As in the proof of Proposition 4.1 (ii), let $j^* \in \operatorname{argmax}_{i \neq i^*} \mu_i$ be the arm with second largest post-click value and define $u_{j^*}^* := u(s^*(\mu_{j^*}), \mu_{j^*})$. Now, let $s'$ be the largest $s \in [0, 1]$ such that

$$u(s', \mu_{i^*}) \geq u_{j^*}^* \text{ and } u(s' - \varepsilon', \mu_{i^*}) > u_{j^*}^* \text{ for all } \varepsilon' > 0.$$

Note that such $s'$ exists since $u$ is continuous and $u(s^*(\mu_{i^*}), \mu_{i^*}) > u_{j^*}^*$. We again distinguish between two cases, similarly to the proof of Proposition 4.1.

**Case 1.** If $u(s', \mu_{i^*}) > u_{j^*}^*$, it follows that $s' = 1$. This means that $\boldsymbol{s} = (s_{i^*}, s_{-i^*})$ with $s_{i^*} = 1$ and arbitrary $s_{-i^*} \in [0, 1]^{K-1}$ form a pure strategy Nash equilibrium for the arms. As in the proof of (i), we then obtain

$$u(s^*, \mu^*) - u(s_{i^*}, \mu_{i^*}) = \beta$$

which implies order $\Omega(\beta T)$ regret under $(s_{i^*}, s_{-i^*})$.

**Case 2.** Now, suppose that $u(s', \mu_{i^*}) = u_{j^*}^*$. Let $s_{i^*} = s' - \varepsilon'$ and $s_i = s^*(\mu_i)$ for all $i \neq i^*$. We see that $(s_{i^*}, s_{-i^*})$ is a $(T\varepsilon')$-Nash equilibrium under the oracle algorithm. Hence, for any $\varepsilon > 0$, the strategy profile $\boldsymbol{s}_{\varepsilon'} := (s' - \varepsilon', s_{-i^*})$ is a $\varepsilon$-Nash equilibrium for all $\varepsilon' < \frac{\varepsilon}{T}$. Using that $u$ is $L$-Lipschitz, we have

$$|u(s' - \varepsilon', \mu_{i^*}) - u(s_{j^*}, \mu_{j^*})| = |u(s' - \varepsilon', \mu_{i^*}) - u(s', \mu_{i^*})| \leq L\varepsilon',$$

and it follows that

$$|u(s^*, \mu^*) - u(s' - \varepsilon', \mu_{i^*})| \geq |u(s^*, \mu^*) - u(s_{j^*}, \mu_{j^*})| - L\varepsilon' \geq \eta - L\varepsilon',$$

We can choose $\varepsilon' < \frac{\varepsilon}{T}$ sufficiently small so that $L\varepsilon' < 1/T$. Hence, over $T$ rounds the oracle algorithm suffers $\Omega(\eta T)$ regret under the $\varepsilon$-Nash equilibrium given by $\boldsymbol{s}_{\varepsilon'}$. This yields the claimed lower bound.

$\square$

## B  PROOF OF LEMMA 5.1

***Proof of Lemma 5.1.*** We use Glicksberg's theorem Glicksberg (1952), which guarantees the existence of a Nash equilibrium in continuous games with compact strategy space and continuous utility functions $v_i$. The strategy space $[0,1]$ is compact and we are left with proving the continuity of $v_i(\text{UCB-S}, \boldsymbol{s})$ in $\boldsymbol{s} \in [0,1]^K$. Since $v_i(\text{UCB-S}, \boldsymbol{s}) = \mathbb{E}_{\boldsymbol{s}}[n_T(i)]s_i$, the question is whether $\mathbb{E}_{\boldsymbol{s}}[n_T(i)]$ is continuous in $\boldsymbol{s}$ under UCB-S. The choice of $\boldsymbol{s}$ influences the actions of UCB-S when through the screening rule in line 10, but also the UCB-type selection in line 4, since post-click rewards are only observed when the arm is clicked.

Let $\mathcal{H}_t$ denote the history of the mechanism's selections and observations up to round $t$, consisting of tuples $(i_t, c_{t,i_t}, r_{t,i_t})$. Even though $r_{t,i_t}$ is sometimes not observed, we include it here and note that it will not matter as the realizations of $r_{t,i_t}$ are independent of $\boldsymbol{s}$. We let $\mathscr{H}_t$ up round $t$ denote the set of all possible histories.

While we are interested in $\mathbb{E}_{\boldsymbol{s}}[n_T(i)]$, for technical reasons, it will be more convenient to prove the continuity of $\mathbb{P}_{\boldsymbol{s}}(\mathcal{H}_t \in \cdot)$ as a function of $\boldsymbol{s}$. We will do so by induction over $t \in [T]$. Naturally, $\mathbb{P}_{\boldsymbol{s}}(\mathcal{H}_1 \in \cdot)$ is continuous in $\boldsymbol{s}$, since $\mathbb{P}_{\boldsymbol{s}}(c_{1,i_1} = 1) = s_{i_1}$ and we break ties in line 4 independent of $\boldsymbol{s}$. For the proof by induction, let us now assume that $\mathbb{P}_{\boldsymbol{s}}(\mathcal{H}_t \in \cdot)$ is continuous in $\boldsymbol{s}$. Then, for $t+1$ we find that again $\mathbb{P}_{\boldsymbol{s}}(c_{t+1,i_{t+1}} = 1) = 1 - \mathbb{P}_{\boldsymbol{s}}(c_{t+1,i_{t+1}} = 0) = s_{i_{t+1}}$ is continuous in $\boldsymbol{s}$.[6] The interesting part is then whether $\mathbb{P}_{\boldsymbol{s}}(i_{t+1} = i)$ is continuous in $\boldsymbol{s}$.

**Lemma B.1.** *For any event $A$, if $\mathbb{P}_{\boldsymbol{s}}(A \mid \mathcal{H}_t)$ and $\mathbb{P}_{\boldsymbol{s}}(\mathcal{H}_t)$ are continuous in $\boldsymbol{s}$ for all $\mathcal{H}_t \in \mathscr{H}_t$, then $\mathbb{P}_{\boldsymbol{s}}(A)$ is also continuous in $\boldsymbol{s}$.*

*Proof.* This follows from the law of total probability. $\square$

We begin by analyzing the dependence of the screening rule in line 10 on $\boldsymbol{s}$. First of all, note that for all $i \in A_{t-1} \setminus \{i_t\}$, we always have $i \in A_t$, i.e., no other arm than $i_t$ will ever be eliminated at the end of round $t$. Moreover, since $\mathbb{P}_{\boldsymbol{s}}(c_{t,i_t} = 1) = s_{i_t}$ is continuous in $\boldsymbol{s}$, it follows that $\mathbb{P}_{\boldsymbol{s}}(\underline{s}_{i_t}^t > a)$ and $\mathbb{P}_{\boldsymbol{s}}(\underline{s}_{i_t}^t > a \mid \mathcal{H}_t)$ are also continuous in $\boldsymbol{s}$ for all $a \in \mathbb{R}$. Consequently, the probability that arm $i_t$ is eliminated in line 10 at the end of round $t$, i.e., $\mathbb{P}_{\boldsymbol{s}}(i_t \notin A_t)$, must be continuous in $\boldsymbol{s}$.

Let us assume that $A_t \neq \emptyset$, since $\mathbb{P}_{\boldsymbol{s}}(i_{t+1} = i)$ is always continuous in $\boldsymbol{s}$ if $A_t = \emptyset$. If $i \notin A_t$, we have $\mathbb{P}_{\boldsymbol{s}}(i_{t+1} = i) = 0$. Note that for all $i \neq i_t$, we have $\overline{\mu}_i^t = \overline{\mu}_i^{t-1}$. We will now first consider any $i \neq i_t$. If $i \in A_t$, we then have

$$
\mathbb{P}_{\boldsymbol{s}}(i_{t+1} = i \mid \mathcal{H}_t)
$$
$$
= \mathbb{P}_{\boldsymbol{s}}\big(\overline{\mu}_i^t > \max_{j \in A_t \setminus \{i,i_t\}} \overline{\mu}_j^t \mid i_t \notin A_t, \mathcal{H}_t\big) \cdot \mathbb{P}_{\boldsymbol{s}}(i_t \notin A_t \mid \mathcal{H}_t)
$$
$$
+ \mathbb{P}_{\boldsymbol{s}}\big(\overline{\mu}_i^t > \max_{j \in A_t \setminus \{i\}} \overline{\mu}_j^t \mid i_t \in A_t, \mathcal{H}_t\big) \cdot \mathbb{P}_{\boldsymbol{s}}(i_t \in A_t \mid \mathcal{H}_t) \tag{4}
$$
$$
= \mathbb{P}\big(\overline{\mu}_i^{t-1} > \max_{j \in A_t \setminus \{i,i_t\}} \overline{\mu}_j^{t-1} \mid i_t \notin A_t, \mathcal{H}_t\big) \cdot \mathbb{P}_{\boldsymbol{s}}(i_t \notin A_t \mid \mathcal{H}_t)
$$
$$
+ \mathbb{P}\big(\overline{\mu}_i^{t-1} > \max_{j \in A_t \setminus \{i,i_t\}} \overline{\mu}_j^{t-1} \mid i_{t+1} \neq i_t, \mathcal{H}_t\big) \cdot \mathbb{P}_{\boldsymbol{s}}(i_{t+1} \neq i_t \mid i_t \in A_t, \mathcal{H}_t) \cdot \mathbb{P}_{\boldsymbol{s}}(i_t \in A_t \mid \mathcal{H}_t).
$$

The leading factors are independent of $\boldsymbol{s}$ and we have already shown that $\mathbb{P}_{\boldsymbol{s}}(i_t \notin A_{t+1})$ is continuous in $\boldsymbol{s}$. We are thus left with proving the continuity of $\mathbb{P}_{\boldsymbol{s}}(i_{t+1} \neq i_t \mid i_t \in A_{t+1}, \mathcal{H}_t)$.

It holds that $\mathbb{P}_{\boldsymbol{s}}(\overline{\mu}_{i_t}^t \in \cdot \mid \mathcal{H}_t) = s_{i_t}\mathbb{P}(\overline{\mu}_{i_t}^t \in \cdot \mid c_{t,i_t} = 1, \mathcal{H}_t) + (1 - s_{i_t})\mathbb{P}(\overline{\mu}_{i_t}^t \in \cdot \mid c_{t,i_t} = 0, \mathcal{H}_t)$, where we used that $\mathbb{P}_{\boldsymbol{s}}(\overline{\mu}_{i_t}^t \in \cdot \mid c_{t,i_t}, \mathcal{H}_t) = \mathbb{P}(\overline{\mu}_{i_t}^t \in \cdot \mid c_{t,i_t}, \mathcal{H}_t)$ is independent of $\boldsymbol{s}$ (conditional on the click-event $c_{t,i_t}$). Hence, as a sum and product of continuous functions $\mathbb{P}_{\boldsymbol{s}}(\overline{\mu}_{i_t}^t \in \cdot \mid \mathcal{H}_t)$ is continuous in $\boldsymbol{s}$ and we get that

$$
\mathbb{P}_{\boldsymbol{s}}(i_{t+1} = i_t \mid \mathcal{H}_t) = \mathbb{P}_{\boldsymbol{s}}(\overline{\mu}_{i_t}^t > \max_{j \neq i_t} \overline{\mu}_j^{t-1} \mid \mathcal{H}_t)\, \mathbb{P}_{\boldsymbol{s}}(i_t \in A_{t+1} \mid \mathcal{H}_t)
$$

---

[6] Note that $r_{t+1,i_{t+1}}$ is independent of $\boldsymbol{s}$.

is continuous in $s$, where we used that $\overline{\mu}_j^t = \overline{\mu}_j^{t-1}$ for all $j \neq i_t$ independent of $s$.[7] Then, since $\mathbb{P}_s(i_{t+1} = i_t \mid i_t \notin A_t) = 0$, we have

$$\mathbb{P}_s(i_{t+1} = i_t \mid \mathcal{H}_t) = \mathbb{P}_s(i_{t+1} = i_t \mid i_t \in A_t, \mathcal{H}_t)\, \mathbb{P}_s(i_t \in A_t \mid \mathcal{H}_t),$$

which shows the continuity of $\mathbb{P}_s(i_{t+1} = i_t \mid i_t \in A_t, \mathcal{H}_t)$. Hence, in view of equation (4), we obtain that $\mathbb{P}_s(i_{t+1} = i \mid \mathcal{H}_t)$ is continuous in $s$. Finally, Lemma B.1 tells us that, since $\mathbb{P}_s(\mathcal{H}_t)$ is assumed to be continuous, $\mathbb{P}_s(i_{t+1} = i)$ is continuous as well. Hence, $\mathbb{P}_s(\mathcal{H}_{t+1})$ is continuous and by induction we get that $\mathbb{P}_s(\mathcal{H}_T)$ is continuous, which implies the continuity of $\mathbb{E}_s[n_T(i)]$ in $s$ for all $i$.

$\square$

## C  PROOF OF THEOREM 5.2

***Proof of Theorem 5.2.*** In the following let $\boldsymbol{\sigma} = (\sigma_1, \ldots, \sigma_K) \in \mathrm{NE}(\text{UCB-S})$ and $s \in \mathrm{supp}(\boldsymbol{\sigma})$. We start of with some preliminaries. Recall that the arm $i$'s utility function given algorithm UCB-S and strategies $s = (s_i, s_{-i})$ can be expressed as

$$v_i(\text{UCB-S}, s_i, s_{-i}) = \mathbb{E}_{(\text{UCB-S}, s_i, s_{-i})}[n_T(i)]s_i,$$

and $v_i(\text{UCB-S}, s_i, \sigma_{-i}) = \mathbb{E}_{s_{-i} \sim \sigma_{-i}}[v_i(\text{UCB-S}, s_i, s_{-i})] = \mathbb{E}_{(s_i, \sigma_{-i})}[n_T(i)]s_i$. For convenience, we omit the argument UCB-S in the following, as every probability and expectation will be w.r.t. UCB-S. The following variables will prove useful. Let $\tau_i$ be the first round that arm $i$ is not in the active set $A_t$ anymore,

$$\tau_i := \min\{t \in [T] \colon i \notin A_t\},$$

and let $\tau$ be the first rounds in which $A_t$ is empty,

$$\tau := \min\{t \in [T] \colon A_t = \emptyset\}.$$

Here, we introduce the convention that $\tau_i = T$ if $i \in A_T$ and $\tau = T$ if $A_T \neq \emptyset$.

To characterize the strategy profiles in the support of any Nash equilibrium under UCB-S, we are going to rely on the best response property of the Nash equilibrium. More precisely, for any $s \in \mathrm{supp}(\boldsymbol{\sigma})$ with $\boldsymbol{\sigma} \in \mathrm{NE}(\text{UCB-S})$ arm $i$'s strategy, $s_i$, must be a best response to $\sigma_{-i}$, i.e., for all $s_i' \in [0, 1]$:

$$v_i(s_i, \sigma_{-i}) \geq v_i(s_i', \sigma_{-i}).$$

In a first step, we show that UCB-S incentivizes arms to choose strategies $s_i$ at least as large as the desired strategy $s^*(\mu_i)$. While this seems obvious at first since each arm $i$'s utility includes a linear factor of $s_i$, we notice that in the click-bandit model arms can prevent the principal from learning about their true post-click value $\mu$ by choosing low click-rates $s$. This could in theory be a viable strategy for suboptimal arms, i.e., $\mu_i < \mu^*$, since it would delay the principal from detecting that the arm is suboptimal. However, we quickly notice that delaying UCB-S from learning about $\mu_1, \ldots, \mu_K$ is to each arm's disadvantage as any delay simply delays the round in which it receives utility. Moreover, while an arm may delay the learning of $\mu_i$, UCB-S still improves its estimate of $s_i$ and the threat of elimination becomes more imminent.

**Lemma C.1.** *For all $s \in \mathrm{supp}(\boldsymbol{\sigma})$ with $\boldsymbol{\sigma} \in \mathrm{NE}(\text{UCB-S})$ and all $i \in [K]$:*

$$s_i \geq s^*(\mu_i).$$

*Proof.* Let $\boldsymbol{\sigma} \in \mathrm{NE}(\text{UCB-S})$. We begin by making some fundamental observations about UCB-S in the click-bandit model. Let $t < T$. If $c_{t,i_t} = 0$ and $i_t \in A_t$, then $i_{t+1} = i_t$.[8] To see this, note that

---

[7] Note that $\overline{\mu}_i^{t-1}$ is $\mathcal{H}_t$-measurable for all $i$.

[8] W.l.o.g. we assume that there are no ties (ignoring the rounds where no post-click rewards have yet been observed). In fact, when there is a possibility of a tie, it can be seen that the arms have an even larger incentive to choose $s_i \geq s^*(\mu_i)$, since they are not guaranteed to be pulled again in the ensuing round when not clicked.

the estimates of $\mu_1, \ldots, \mu_K$ and their confidence bounds do not change from $t$ to $t+1$ if $c_{t,i_t} = 0$, since no post-click reward was observed for any of the arms. Hence, given that $i_t \in A_t$, we have

$$i_{t+1} = \operatorname*{argmax}_{i \in A_t} \overline{\mu}_i^t = \operatorname*{argmax}_{i \in A_{t-1}} \overline{\mu}_i^{t-1} = i_t.$$

Thus, given that $i_t$ is not eliminated in the mean time, UCB-S plays arm $i_t$ until arm $i_t$ is clicked, i.e., until the arm receives utility 1, or we've reached round $T$. Hence, whenever $c_{t,i_t} = 0$, it simply delays the UCB selection rule by one round as the estimates and confidences of $\mu_1, \ldots, \mu_K$ do not change. At the same time, arm $i$ with $i = i_t$ still only receives utility 1 for this sequence of selections by UCB-S, since the UCB selection rule "progresses" once $c_{t,i_t} = 1$.

More formally, we can define the phases of the UCB selection rule recursively by $\eta_k := \min\{t > \eta_{k-1} : c_{t,i_t} = 1\}$ with $\eta_0 := 0$ and $\eta_k = \infty$ if round $T$ is exceeded without a click. We define the number of such rounds as $N := \max\{k : \eta_k < \infty\}$ and remark that $N \leq T$ always.

We first note that conditional on $A_{\eta_{k-1}}$ the identity of $i_{\eta_k}$ is independent of $s_i$ (and $\sigma_{-i}$), but only depends on $\mu_1, \ldots, \mu_K$ and their realization at rounds $\eta_1, \ldots, \eta_{k-1}$, i.e., $\mathbb{P}_{(s_i, \sigma_{-i})}(i_{\eta_k} = i \mid A_{\eta_{k-1}}) = \mathbb{P}(i_{\eta_k} = i \mid A_{\eta_{k-1}})$. Moreover, we also see that $\mathbb{P}_{(s_i, \sigma_{-i})}(A_{\eta_k} = \cdot \mid i \in A_{\eta_k})$ is independent of $s_i$.[9] Then, since

$$\mathbb{P}_{(s_i, \sigma_{-i})}(i_{\eta_k} = i \mid i \in A_{\eta_{k-1}})$$
$$= \sum_A \mathbb{P}_{(s_i, \sigma_{-i})}(i_{\eta_k} = i \mid A_{\eta_{k-1}} = A \cup \{i\}) \, \mathbb{P}(A_{\eta_{k-1}} = A \mid i \in A_{\eta_{k-1}}),$$

this implies that $\mathbb{P}_{(s_i, \sigma_{-i})}(i_{\eta_k} = i \mid i \in A_{\eta_{k-1}})$ is independent of $s_i$. Using the shown independence, let us then write

$$\mathbb{P}_{(s_i, \sigma_{-i})}(i_{\eta_k} = i) = \mathbb{P}(i_{\eta_k} = i \mid i \in A_{\eta_{k-1}}) \, \mathbb{P}_{(s_i, \sigma_{-i})}(i \in A_{\eta_{k-1}})$$
$$+ \mathbb{P}_{(s_i, \sigma_{-i})}(i_{\eta_k} = i \mid i \notin A_{\eta_{k-1}}) \, \mathbb{P}_{(s_i, \sigma_{-i})}(i \notin A_{\eta_{k-1}}). \tag{5}$$

Now, it holds that $\mathbb{P}(i_{\eta_k} = i \mid i \in A_{\eta_{k-1}}) \geq \mathbb{P}_{(s_i, \sigma_{-i})}(i_{\eta_k} = i \mid i \notin A_{\eta_{k-1}})$ always. Naturally, for $s_i < s^*(\mu_i)$ we have $\mathbb{P}_{(s_i, \sigma_{-i})}(i \in A_{\eta_{k-1}}) \leq \mathbb{P}_{(s^*(\mu_i), \sigma_{-i})}(i \in A_{\eta_{k-1}})$ so that from equation (5) it follows that

$$\mathbb{P}_{(s_i, \sigma_{-i})}(i_{\eta_k} = i) \leq \mathbb{P}_{(s^*(\mu_i), \sigma_{-i})}(i_{\eta_k} = i). \tag{6}$$

We also see that as $s_i$ decreases the number of utility-yielding rounds decreases in expectation, i.e., for $s_i < s^*(\mu_i)$:

$$\mathbb{E}_{(s_i, \sigma_{-i})}[N] < \mathbb{E}_{(s^*(\mu_i), \sigma_{-i})}[N] \tag{7}$$

since $\eta_k - \eta_{k-1} \sim \mathrm{Geom}(s_{i_{\eta_k}})$. Finally, it follows from equations (6) and (7) and a technical lemma about the comparison of expectation under two measures (Lemma G.1 in Appendix G) that

$$\mathbb{E}_{(s_i, \sigma_{-i})}[m_T(i)] = \mathbb{E}_{(s_i, \sigma_{-i})}\left[\sum_{k=1}^N \mathbb{1}_{\{i_{\eta_k} = i\}}\right]$$
$$< \mathbb{E}_{(s^*(\mu_i), \sigma_{-i})}\left[\sum_{k=1}^N \mathbb{1}_{\{i_{\eta_k} = i\}}\right] = \mathbb{E}_{(s^*(\mu_i), \sigma_{-i})}[m_T(i)].$$

Since a post-click reward is observed with probability $s_i$ every time an arm is pulled by the learner, we have $\mathbb{E}_{(s_i, \sigma_{-i})}[m_t(i)] = \mathbb{E}_{(s_i, \sigma_{-i})}[n_t(i)] s_i$ so that $v_i(s_i, \sigma_{-i}) = \mathbb{E}_{(s_i, \sigma_{-i})}[m_t(i)]$. Now, from the above we see that for any $s_i < s^*(\mu_i)$, the strategy $s^*(\mu_i)$ is a strictly better response to $\sigma_{-i}$ than $s_i$, i.e., $v_i(s^*(\mu_i), \sigma_{-i}) > v_i(s_i, \sigma_{-i})$. This shows that $s_i \geq s^*(\mu_i)$ for any $s_i \in \mathrm{supp}(\sigma_i)$ with $\boldsymbol{\sigma} \in \mathrm{NE}(\text{UCB-S})$.

$\square$

---

[9]However, note that the value of $A_t$ for general $t$ is not independent of $s_i$ conditional on $i \in A_t$, since, e.g., for small $s_i$ other arms will be played fewer times before round $t$, thereby reducing the probability of them being eliminated by round $t$.

We continue the proof of Theorem 5.2 by decomposing the number of times each arm is selected by UCB-S. Given $(s_i, \sigma_{-i})$ we can split $\mathbb{E}_{(s_i, \sigma_{-i})}[n_T(i)]$ into the time steps before $\tau_i$ and after $\tau$, since arm $i$ is never played in the rounds between $\tau_i$ and $\tau$. Recall that UCB-S plays arms uniformly at random after round $\tau$ so that

$$\mathbb{E}_{(s_i, \sigma_{-i})}[n_T(i)] = \mathbb{E}_{(s_i, \sigma_{-i})} \left[ \sum_{t=1}^{\tau_i} \mathbb{1}_{\{i_t = i\}} + \sum_{t=\tau+1}^{T} \mathbb{1}_{\{i_t = i\}} \right]$$

$$= \mathbb{E}_{(s_i, \sigma_{-i})}[n_{\tau_i}(i)] + \mathbb{E}_{(s_i, \sigma_{-i})}\left[ \frac{T - \tau}{K} \right]. \tag{8}$$

The proof of Theorem 5.2 proceeds by upper and lower bounding the quantities in (8), which will eventually lead to an approximate characterization of the best response $s_i$. More precisely, we establish the following bounds for $\sigma \in \text{NE(UCB-S)}$ and $s_i \in \text{supp}(\sigma_i)$:

Lemma C.2: $\mathbb{E}_{(s_i, \sigma_{-i})}[n_{\tau_i}(i)] \leq \mathcal{O}\left( \frac{H^2 \log(T)}{s_i(s_i - s^*(\mu_i))^2} \right)$.

Lemma C.4: $T - \mathbb{E}_{(s_i, \sigma_{-i})}[\tau] \leq \mathcal{O}(1)$.

Lemma C.5: $\mathbb{E}_{(s_i, \sigma_{-i})}[n_T(i)] = \Omega\left( \min\{ \frac{\log(T)}{s_i \Delta_i^2}, s^*(\mu_i)\frac{T}{K} \} \right)$.

## C.1 Bounds on $n_T(i)$, $n_{\tau_i}(i)$, $\tau_i$, and $\tau$ under UCB-S

We begin by bounding the number of allocations arm $i$ receives before elimination. As one expects, UCB-S is able to detect that $s_i \neq s^*(\mu_i)$ with high probability after at most $\mathcal{O}(1/(s_i - s^*(\mu_i))^2)$ selections.

**Lemma C.2.** *Let $\sigma \in \text{NE(UCB-S)}$ and $s_i \in \text{supp}(\sigma_i)$ with $s_i \neq s^*(\mu_i)$. Then, the number of times that $i$ is being selected before elimination, $n_{\tau_i)}(i)$, satisfies the following. For some constant $c_1 > 0$, it holds that*

$$\mathbb{P}_{(s_i, \sigma_{-i})}\left( n_{\tau_i}(i) \leq c_1 \frac{H^2 \log(T)}{s_i(s_i - s^*(\mu_i))^2} \right) \geq 1 - \frac{3}{T^2},$$

*and as an immediate consequence for some $c_2 > 0$:*

$$\mathbb{E}_{(s_i, \sigma_{-i})}[n_{\tau_i}(i)] \leq c_2 \frac{H^2 \log(T)}{s_i(s_i - s^*(\mu_i))^2}.$$

*Proof.* For simplicity, we consider w.l.o.g. the one-sided elimination rule checking whether the arm $i$'s strategy $s_i$ exceeds the desired strategy $s^*(\mu_i)$:

$$\underline{s}_i^t > \max_{\mu \in [\underline{\mu}_i^t, \overline{\mu}_i^t]} s^*(\mu). \tag{9}$$

Let $\alpha_t(i) = \sqrt{\frac{2\log(T)}{n_t(i)}}$ and $\beta_t(i) = \sqrt{\frac{2\log(T)}{m_t(i)}}$. Recall that $s^*(\mu)$ is $H$-Lipschitz. Then,

$$\max_{\mu \in [\underline{\mu}_i^t, \overline{\mu}_i^t]} s^*(\mu) \leq s^*(\widehat{\mu}_i^t) + H\beta_t(i).$$

As a consequence, we see that a sufficient condition for the elimination rule (9) to trigger is given by

$$\widehat{s}_i^t - \alpha_t(i) > s^*(\widehat{\mu}_i^t) + H\beta_t(i), \tag{10}$$

where by definition $\underline{s}_i^t = \widehat{s}_i^t - \alpha_t(i)$. The following statements are always w.r.t. $(s_i, \sigma_{-i})$, i.e., w.r.t. the probability measure $\mathbb{P}_{(s_i, \sigma_{-i})}$. From Hoeffding's inequality, we know that with probability at least $1 - 1/T^2$:

$$|\widehat{s}_i^t - s_i| \leq \alpha_t(i).$$

Similarly, using the Lipschitzness of $s^*(\mu)$, Hoeffding's inequality implies that with probability at least $1 - 1/T^2$:

$$|s^*(\widehat{\mu}_i^t) - s^*(\mu_i)| \le H |\widehat{\mu}_i^t - \mu_i| \le H\beta_t(i).$$

It then follows that with probability at least $1 - 2/T^2$

$$\widehat{s}_i^t - s^*(\widehat{\mu}_i^t) \ge \big(s_i - s^*(\mu_i)\big) - \big(\alpha_t(i) + \beta_t(i)\big) \ge \big(s_i - s^*(\mu_i)\big) - (H+1)\beta_t(i),$$

where we used that $\alpha_t(i) = \sqrt{\frac{2\log(T)}{n_t(i)}} \le \sqrt{\frac{2\log(T)}{m_t(i)}} = \beta_t(i)$, since $n_t(i) > m_t(i)$ by definition. Therefore, the sufficient condition in equation (10) is satisfied with probability $1 - 2/T^2$ for

$$s_i - s^*(\mu_i) > 2(H+1)\beta_t(i) = 2(H+1)\sqrt{\frac{2\log(T)}{m_t(i)}}.$$

In other words, arm $i$ has been eliminated by round $t$ with probability at least $1 - 2/T^2$ if

$$m_t(i) > \frac{16H^2\log(T)}{(s_i - s^*(\mu_i))^2}. \tag{11}$$

Lastly, we translate this to a statement about $n_t(i)$. Recall that conditional on $n_t(i)$, we have $\mathbb{E}[m_t(i) \mid n_t(i)] = n_t(i)s_i$, since arm $i$ is clicked with probability $s_i$. From Hoeffding's inequality, we then again have with probability $1 - 1/T^2$

$$|m_t(i) - n_t(i)s_i| \le \sqrt{2n_t(i)\log(T)}$$

and thus $m_t(i) \ge n_t(i)s_i - \sqrt{2n_t(i)\log(T)}$. Then, in view of equation (11), if

$$n_t(i) > c_1 \frac{H^2\log(T)}{s_i(s_i - s^*(\mu_i))^2}$$

for some sufficiently large $c_2 > 0$, then with probability at least $1 - 3/T^2$ arm $i$ has been eliminated before round $t$. Since $\tau_i$ denotes the round in which $i$ is eliminated from $A_t$, this means that with probability $1 - 3/T^2$:

$$n_{\tau_i}(i) \le c_1 \frac{H^2\log(T)}{s_i(s_i - s^*(\mu_i))^2}.$$

Since by definition $\tau_i \le T$, this implies that for some $c_2 > 0$:

$$\mathbb{E}_{(s_i,\sigma_{-i})}[n_{\tau_i}(i)] \le c_2 \frac{H^2\log(T)}{s_i(s_i - s^*(\mu_i))^2}.$$

$\square$

We briefly recall a standard result often used in the context of MABs, which states that any probably correct decision rule needs $\Omega(\frac{1}{\varepsilon^2})$ samples to distinguish between two hypotheses for which the Bernoulli means lie $\varepsilon$ apart. We only give a short outline of the proof and refer to the many expositions of such bounds for more detail (see, e.g., Theorem 1 in (Mannor and Tsitsiklis, 2004), Section 2 in (Slivkins et al., 2019), Section 14 in (Lattimore and Szepesvári, 2020)).

**Lemma C.3.** *In order for us to reuse our current notation, suppose that $K = 1$. In this case, $n_{\tau_1}(1)$ simply denotes the number of samples from arm 1 before it gets eliminated, i.e., UCB-S asserts that $s_1 \ne s^*(\mu_1)$. For $s_1 \ne s^*(\mu_1)$, it holds that*

$$\mathbb{E}_{s_1}[n_{\tau_1}(1)] \ge \Omega\left(\frac{\log(T)}{(s_1 - s^*(\mu_1))^2}\right).$$

*Proof.* W.l.o.g. we can assume that $r_{t,1} = \mu_1$ for all $t$ so that we are only concerned with the estimation of the Bernoulli mean $s_1$ (this clearly only reduces the number of samples the elimination rule would need). Note that the elimination rule is correct with probability $1 - 1/T^2$ by construction

of the confidence sets around $s_1$, i.e., only eliminates arm 1 if it in fact deviated from $s^*(\mu_1)$. We can then consider the hypotheses

$$H_0 : s_1 = s^*(\mu_1) \quad \text{and} \quad H_1 : s_1 = s^*(\mu_1) + \varepsilon.$$

Then, since the elimination rule is correct with probability $1 - 1/T^2$, the standard hypothesis testing argument (see, e.g., Theorem 1 in (Mannor and Tsitsiklis, 2004)) yields for some constant $c > 0$ that $\mathbb{E}_{s_1}[n_{\tau_1}(1)] \geq \frac{c \log(T)}{\varepsilon^2} = \frac{c \log(T)}{(s_1 - s^*(\mu_1))^2}$.

$\square$

The next lemma states that $\mathbb{E}_{(s_i, \sigma_{-i})}[\tau]$ is close to $T$. The intuition of this is quickly explained. If the set $A_t$ becomes empty, UCB-S plays arms uniformly at random. However, if one arm would happen to remain in $A_t$ this arm would always be played (as it has no competition). To do so, an arm simply has to ensure that it does not get eliminated too early. Now, in view of Lemma C.3, an arm can be sampled order $x$ more times without getting eliminated for moving its strategy order $\sqrt{x}$ closer to $s^*(\mu_i)$. Writing the arms' utility as $v_i(s_i, \sigma_{-i}) = \mathbb{E}_{(s_i, \sigma_{-i})}[n_T(i)]s_i = \mathbb{E}_{(s_i, \sigma_{-i})}[n_T(i)]\big(s^*(\mu_i) + (s_i - s^*(\mu_i))\big)$ we see that a quadratic increase in $\mathbb{E}_{(s_i, \sigma_{-i})}[n_T(i)]$ will dominate a linear decrease in $s_i - s^*(\mu_i)$.

**Lemma C.4.** *Let $\sigma \in \text{NE(UCB-S)}$ and $s_i \in \text{supp}(\sigma_i)$. Then,*

$$\mathbb{E}_{(s_i, \sigma_{-i})}[\tau] \geq T - \mathcal{O}(1).$$

*Proof.* Let $s^*(\mu_i) \leq s_i' < s_i$. Due to delays for smaller click-rates (see proof of Lemma C.1) and the fact that under $s_i'$ the probability of arm $i$ being eliminated at any given round is smaller than under $s_i$, it holds for all $j \neq i$ that

$$\mathbb{E}_{(s_i', \sigma_{-i})}[n_{\tau_j}(j)] \leq \mathbb{E}_{(s_i, \sigma_{-i})}[n_{\tau_j}(j)].$$

By definition of $\tau$, we have $\mathbb{E}_{(s_i, \sigma_{-i})}[\tau] = \sum_{j \in [K]} \mathbb{E}_{(s_i, \sigma_{-i})}[n_{\tau_j}(j)]$ so that the above implies

$$\sum_{j \neq i} \mathbb{E}_{(s_i', \sigma_{-i})}[n_{\tau_j}(j)] \leq \mathbb{E}_{(s_i, \sigma_{-i})}[n_{\tau_j}(j)] = \mathbb{E}_{(s_i, \sigma_{-i})}[\tau] - \mathbb{E}_{(s_i, \sigma_{-i})}[n_{\tau_i}(i)].$$

In other words, under any strategy $s^*(\mu_i) \leq s_i' < s_i$, all arms $j \neq i$ will be eliminated after a total of $\mathbb{E}_{(s_i, \sigma_{-i})}[\tau] - \mathbb{E}_{(s_i, \sigma_{-i})}[n_{\tau_i}(i)]$ rounds so that there are at least $\mathbb{E}_{(s_i, \sigma_{-i})}[n_{\tau_i}(i)] + T - \mathbb{E}_{(s_i, \sigma_{-i})}[\tau]$ many "uncontested" rounds.

For convenience, let $N(s_i, \sigma_{-i}) = T - \mathbb{E}_{(s_i, \sigma_{-i})}[\tau]$, i.e., the expected number of rounds that $A_t$ is empty and arms are being selected uniformly at random. Now, in view of Lemma C.3, there exists $s_i'$ with

$$s_i' - s^*(\mu_i) \geq \Omega\left(\sqrt{\frac{\log(T)}{\mathbb{E}_{(s_i, \sigma_{-i})}[n_{\tau_i}(i)] + N(s_i, \sigma_{-i})}}\right)$$

such that $\mathbb{E}_{(s_i', \sigma_{-i})}[n_{\tau_i}(i)] \geq \mathbb{E}_{(s_i, \sigma_{-i})}[n_{\tau_i}(i)] + N(s_i, \sigma_{-i})$.

The proof proceeds by contradiction. To this end, suppose the contrary is true, namely, that $N(s_i, \sigma_{-i})$ is not constant, but in fact increasing in $T$, i.e., $N(s_i, \sigma_{-i}) = w(1)$. We then show that $v_i(s_i', \sigma_{-i}) > v_i(s_i, \sigma_{-i})$, which is a contradiction to $s_i$ being a best response to $\sigma_{-i}$. From Lemma C.2 we know that

$$s_i - s^*(\mu_i) \leq \mathcal{O}\left(\sqrt{\frac{\log(T)}{s^*(\mu_i)\mathbb{E}_{(s_i, \sigma_{-i})}[n_{\tau_i}(i)]}}\right),$$

where we used that $s_i \geq s^*(\mu_i)$ by Lemma C.1. Using that $\mathbb{E}_{(s'_i, \sigma_{-i})}[n_T(i)] \geq \mathbb{E}_{(s'_i, \sigma_{-i})}[n_{\tau_i}(i)]$, we then obtain

$$
\begin{aligned}
v_i(s'_i, \sigma_{-i}) = &\, \mathbb{E}_{(s'_i, \sigma_{-i})}[n_T(i)]s'_i \\
\geq &\, \mathbb{E}_{(s'_i, \sigma_{-i})}[n_{\tau_i}(i)]\big(s^*(\mu_i) + (s'_i - s^*(\mu_i))\big) \\
\geq &\, \big(\mathbb{E}_{(s_i, \sigma_{-i})}[n_{\tau_i}(i)] + N(s_i, \sigma_{-i})\big)\left(s^*(\mu_i) + \Omega\left(\sqrt{\frac{\log(T)}{\mathbb{E}_{(s_i, \sigma_{-i})}[n_{\tau_i}(i)] + N(s_i, \sigma_{-i})}}\right)\right) \\
\geq &\, \big(\mathbb{E}_{(s_i, \sigma_{-i})}[n_{\tau_i}(i)] + N(s_i, \sigma_{-i})\big)s^*(\mu_i) + \Omega\left(\sqrt{\log(T)\big(\mathbb{E}_{(s_i, \sigma_{-i})}[n_{\tau_i}(i)] + N(s_i, \sigma_{-i})\big)}\right) \\
> &\, \mathbb{E}_{(s_i, \sigma_{-i})}[n_{\tau_i}(i)]s^*(\mu_i) + \frac{2N(s_i, \sigma_{-i})}{K}s^*(\mu_i) + \mathcal{O}\left(\sqrt{\log(T)\mathbb{E}_{(s_i, \sigma_{-i})}[n_{\tau_i}(i)]}\right) \\
\geq &\, \big(\mathbb{E}_{(s_i, \sigma_{-i})}[n_{\tau_i}(i)] + \frac{N(s_i, \sigma_{-i})}{K}\big)\big(s_i + (s_i - s^*(\mu_i))\big) \\
\geq &\, \mathbb{E}_{(s_i, \sigma_{-i})}[n_T(i)]s_i \\
= &\, v_i(s_i, \sigma_{-i}).
\end{aligned}
$$

Hence, $s'_i$ is a better response to $\sigma_{-i}$ than $s_i$, which is a contradiction to $s_i \in \text{supp}(\sigma_i)$.

$\square$

The next lemma *lower bounds* $\mathbb{E}_{(s_i, \sigma_{-i})}[n_T(i)]$ for which we distinguish between optimal and sub-optimal arms in terms of post-click rewards $\mu$.

**Lemma C.5.** *Let $\sigma \in \text{NE(UCB-S)}$.*

*(i) For all $i^* \in [K]$ with $\Delta_{i^*} = 0$ and $s_{i^*} \in \text{supp}(\sigma_{i^*})$:*

$$
\mathbb{E}_{(s_{i^*}, \sigma_{-i^*})}[n_T(i^*)] \geq s^*(\mu_{i^*})\Omega\left(\frac{T}{K}\right).
$$

*(ii) For all $i \in [K]$ with $\Delta_i > 0$ and $s_i \in \text{supp}(\sigma_i)$:*

$$
\mathbb{E}_{(s_i, \sigma_{-i})}[n_T(i)] \geq \Omega\left(\min\left\{\frac{\log(T)}{s_i\Delta_i^2}, s^*(\mu_i)\frac{T}{K}\right\}\right).
$$

*Proof.* *(i)*: Let $\sigma \in \text{NE(UCB-S)}$ and let $i^* \in [K]$ such that $\Delta_{i^*} = 0$. Recall that when playing strategy $s^*(\mu_{i^*})$ arm $i^*$ is eliminated with low probability so that

$$
\mathbb{P}_{(s^*(\mu_{i^*}), \sigma_{-i^*})}(i^* \in A_T) \geq 1 - 1/T^2.
$$

Now, given that $i^*$ is not going to be eliminated, the UCB-type selection rule of UCB-S selects any arm $i^*$ with maximal post-click reward $\mu_{i^*} = \mu^*$ at least $\Omega(T/K)$ times so that $\mathbb{E}_{(s^*(\mu_{i^*}), \sigma_{-i^*})}[n_T(i^*)] \geq \Omega(T/K)$. Then, since $s_{i^*}$ has to be a best response to $\sigma_{-i^*}$, we obtain

$$
\begin{aligned}
\mathbb{E}_{(s_{i^*}, \sigma_{-i^*})}[n_T(i^*)] \geq &\, s_{i^*}\mathbb{E}_{(s_{i^*}, \sigma_{-i^*})}[n_T(i^*)] = v_{i^*}(s_{i^*}, \sigma_{-i^*}) \\
\geq &\, v_{i^*}(s^*(\mu_{i^*}), \sigma_{-i^*}) \geq s^*(\mu_{i^*})\mathbb{E}_{(s^*(\mu_{i^*}), \sigma_{-i^*})}[n_T(i^*)] \geq s^*(\mu_{i^*})\Omega\left(\frac{T}{K}\right).
\end{aligned}
$$

*(ii)*: Once again, we use the desired strategy $s^*(\mu_i)$ to infer properties of $s_i$. Let us be reminded that under $s^*(\mu_i)$ arm $i$ is eliminated with low probability, i.e.,

$$
\mathbb{P}_{(s^*(\mu_i), \sigma_{-i})}(i \in A_T) \geq 1 - 1/T^2
$$

so that when studying $(s^*(\mu_i), \sigma_{-i})$ the potential elimination of arm $i$ is negligible.

We will argue about $\mathbb{E}_{(s^*(\mu_i),\sigma_{-i})}[n_T(i)]$ via $\mathbb{E}_{(s^*(\mu_i),\sigma_{-i})}[m_T(i)]$. To isolate the rounds in which arms are clicked, i.e., post click-rewards are observed, we will re-use the rounds $\eta_1, \eta_2, \ldots$, which determine the phases of the UCB selection rule (introduced in Lemma C.1). On the rounds $\eta_1, \eta_2, \ldots$, the UCB-selection rule of line 4 is analogous to standard UCB in a MAB. We can then use well-known results from the instance-dependent lower bound analysis of the MAB problem. From Lemma 16.3 in (Lattimore and Szepesvári, 2020) it then follows that for some constant $c_1 > 0$:[10]

$$\mathbb{E}_{(s^*(\mu_i),\sigma_{-i})}[m_T(i)] \geq \frac{\frac{1}{2}\log(T) + \log\left(\frac{c_1\Delta_i}{\sqrt{K}}\right)}{2\Delta_i^2}.$$

We see that this lower bound is only meaningful for sufficiently large $\Delta_i$, as the numerator may become negative for $\Delta_i = \mathcal{O}(\sqrt{K/T})$. For now let us assume that $\Delta_i$ is sufficiently large. Recall that $\mathbb{E}_{(s^*(\mu_i),\sigma_{-i})}[m_T(i)] = \mathbb{E}_{(s^*(\mu_i),\sigma_{-i})}[n_T(i)]s^*(\mu_i)$ as arm $i$ is clicked with probability $s^*(\mu_i)$. Since $s_i$ must be a best response to $\sigma_{-i}$, it must then hold that

$$\mathbb{E}_{(s^*(\mu_i),\sigma_{-i})}[n_T(i)]s_i = v_i(s_i, \sigma_{-i})$$

$$\geq v_i(s^*(\mu_i), \sigma_{-i}) = \mathbb{E}_{(s^*(\mu_i),\sigma_{-i})}[m_T(i)] \geq c_2\frac{\log(T)}{\Delta_i^2}$$

for some $c_2 > 0$. Solving for $\mathbb{E}_{(s^*(\mu_i),\sigma_{-i})}[n_T(i)]$ then yields

$$\mathbb{E}_{(s^*(\mu_i),\sigma_{-i})}[n_T(i)] \geq c_2\frac{\log(T)}{s_i\Delta_i^2}.$$

Next, for $\Delta_i \leq \mathcal{O}(\sqrt{K/T})$ it is well-known that the number of times UCB plays arm $i$ is order at least $\Omega(T/K)$. We then have $\mathbb{E}_{s^*(\mu_i,\sigma_{-i})}[n_T(i)] = \Omega\left(\frac{T}{K}\right)$, so that

$$\mathbb{E}_{(s_i,\sigma_{-i})}[n_T(i)] \geq s_i\mathbb{E}_{(s_i,\sigma_{-i})}[n_T(i)] = v_{i^*}(s_i, \sigma_{-i})$$

$$\geq v_i(s^*(\mu_i), \sigma_{-i}) \geq s^*(\mu_i)\mathbb{E}_{(s^*(\mu_i),\sigma_{-i})}[n_T(i)] \geq s^*(\mu_i)\Omega\left(\frac{T}{K}\right).$$

$\square$

## C.2 CONNECTING THE BOUNDS

Finally, using the lower and upper bound on $\mathbb{E}_{(s_i,\sigma_{-i})}[n_T(i)]$, we obtain the following approximate characterization of the strategies in the Nash equilibrium $\boldsymbol{\sigma} \in \text{NE(UCB-S)}$.

For $i^* \in [K]$ with $\Delta_{i^*} = 0$, it follows from equation (8) and Lemma C.2, Lemma C.4, Lemma C.5 that

$$s^*(\mu_{i^*})\Omega\left(\frac{T}{K}\right) \leq \mathbb{E}_{(s_{i^*},\sigma_{-i^*})}[n_T(i^*)] \leq \mathcal{O}\left(\frac{H^2\log(T)}{s_{i^*}(s_{i^*} - s^*(\mu_{i^*}))^2}\right) + \mathcal{O}\left(\frac{1}{K}\right).$$

Solving for $s_{i^*} - s^*(\mu_{i^*})$, we obtain

$$s_{i^*}\left(s_{i^*} - s^*(\mu_{i^*})\right)^2 \leq \mathcal{O}\left(\frac{H^2 K\log(T)}{T\, s^*(\mu_{i^*})}\right),$$

Finally, using that $s^*(\mu_{i^*}) \leq s_{i^*}$ by Lemma C.1 yields the claimed bound (note that $s^*(\mu)$ is bounded away from zero by assumption (A3))

$$s_{i^*} - s^*(\mu_{i^*}) \leq \mathcal{O}\left(H\sqrt{\frac{K\log(T)}{T\, s^*(\mu_{i^*})^2}}\right).$$

---

[10]We here used that the standard minimax bandit regret of UCB in MABs is bounded as $\widetilde{\mathcal{O}}(\sqrt{KT})$.

For $i \in [K]$ with $\Delta_i > 0$ suppose that $\frac{\log(T)}{s_i \Delta_i^2} \leq s^*(\mu_i) \frac{T}{K}$. Then, we have

$$\Omega\left(\frac{\log(T)}{s_i \Delta_i^2}\right) \leq \mathbb{E}_{(s_i, \sigma_{-i})}[n_T(i^*)] \leq \mathcal{O}\left(\frac{H^2 \log(T)}{s_i(s_i - s^*(\mu_i))^2}\right) + \mathcal{O}\left(\frac{1}{K}\right),$$

which after solving for $s_i - s^*(\mu_i)$ yields

$$s_i - s^*(\mu_i) \leq \mathcal{O}\left(H\Delta_i\right).$$

For $i \in [K]$ with $\frac{\log(T)}{s_i \Delta_i^2} > s^*(\mu_i) \frac{T}{K}$, it follows, analogously to the case of $\Delta_i = 0$, from Lemma C.2, Lemma C.4, and Lemma C.5 that

$$s_i - s^*(\mu_i) \leq \mathcal{O}\left(H\sqrt{\frac{K \log(T)}{T\, s^*(\mu_i)^2}}\right).$$

$\square$

# D  PROOF OF THEOREM 5.3

***Proof of Theorem 5.3.*** Let $\sigma \in \text{NE(UCB-S)}$ and let $i^* \in [K]$ be any arm with $\Delta_{i^*} = 0$. We begin with a standard regret decomposition into the number of times each arm is played and the rounds before $i^*$ is eliminated. It holds that

$$
\begin{aligned}
R_T(\text{UCB-S}, \sigma) &= \mathbb{E}_{s \sim \sigma}\left[\sum_{t=1}^{T} u(s^*, \mu^*) - u(s_{i_t}, \mu_{i_t})\right] \\
&= \mathbb{E}_{s \sim \sigma}\left[\sum_{t=1}^{\tau_{i^*}} u(s^*, \mu^*) - u(s_{i_t}, \mu_{i_t})\right] + \mathbb{E}_{s \sim \sigma}\left[\sum_{t=\tau_{i^*}+1}^{T} u(s^*, \mu^*) - u(s_{i_t}, \mu_{i_t})\right] \\
&\leq \mathbb{E}_{s \sim \sigma}\left[\sum_{i \in [K]} \mathbb{E}_s[n_{\tau_{i^*}}(i)]\big(u(s^*, \mu^*) - u(s_i, \mu_i)\big)\right] + (T - \mathbb{E}_\sigma[\tau_{i^*}]).
\end{aligned}
\tag{12}
$$

From Lemma D.1 below we know that $T - \mathbb{E}_\sigma[\tau_{i^*}] \leq \sqrt{KT}$. We continue to split the arms into two cases. To this end, let $\Delta_i' := \sqrt{\frac{K \log(T)}{T\, s^*(\mu_i)^2}}$ and let $\Delta_*' = \sqrt{\frac{K \log(T)}{T\, s^*(\mu^*)^2}}$. For $i \in [K]$, we distinguish between (a) $\Delta_i \leq \Delta_i'$ and (b) $\Delta_i > \Delta_i'$.

We begin with (a). Recall that $s^* := s^*(\mu^*)$. For the proof we will need one last technicality, namely, that $\Delta_i' \leq 2\Delta_*'$. We here assume that $s^*(\mu^*) > 2H\Delta_i'$.[11] Then, since $|s^*(\mu^*) - s^*(\mu_i)| \leq H\Delta_i \leq H\Delta_i'$, we get

$$\Delta_i' = \frac{1}{s^*(\mu_i)}\sqrt{\frac{K \log(T)}{T}} \leq \frac{1}{s^*(\mu^*) - H\Delta_i'}\sqrt{\frac{K \log(T)}{T}} \leq \frac{2}{s^*(\mu^*)}\sqrt{\frac{K \log(T)}{T}} = 2\Delta_*'.$$

---

[11]Otherwise there is nothing to prove since the regret bound of Theorem 5.3 is of order $T$.

We can now apply Theorem 5.2 to obtain for any $s \in \text{supp}(\boldsymbol{\sigma})$ that

$$\sum_{i:\Delta_i \leq \Delta_i'} \mathbb{E}_s[n_{\tau_{i^*}}(i)]\big(u(s^*, \mu^*) - u(s_i, \mu_i)\big)$$

$$\leq L \sum_{i:\Delta_i \leq \Delta_i'} \mathbb{E}_s[n_{\tau_{i^*}}(i)]\big(|s^*(\mu^*) - s_i| + |\mu^* - \mu_i|\big)$$

$$\leq L \sum_{i:\Delta_i \leq \Delta_i'} \mathbb{E}_s[n_{\tau_{i^*}}(i)] \left(|s^*(\mu^*) - s^*(\mu_i)| + \mathcal{O}\left(H\sqrt{\frac{K\log(T)}{T\,s^*(\mu_i)^2}}\right) + \Delta_i\right)$$

$$\leq L \sum_{i:\Delta_i \leq \Delta_i'} \mathbb{E}_s[n_{\tau_{i^*}}(i)] \left((H+1)\Delta_i + \mathcal{O}\left(H\sqrt{\frac{K\log(T)}{T\,s^*(\mu_i)^2}}\right) + \Delta_i\right)$$

$$\leq L(H+2) \sum_{i:\Delta_i \leq \Delta_i'} \mathbb{E}_s[n_{\tau_{i^*}}(i)]\,\Delta_i' \tag{13}$$

$$\leq 2L(H+2)\Delta_*' \sum_{i:\Delta_i \leq \Delta_i'} \mathbb{E}_s[n_{\tau_{i^*}}(i)]$$

$$\leq LH \cdot \mathcal{O}\left(H\sqrt{\frac{K\log(T)}{T\,s^*(\mu^*)^2}}\right) \sum_{i:\Delta_i \leq \Delta_i'} \mathbb{E}_s[n_{\tau_{i^*}}(i)]$$

$$\leq \frac{LH}{s^*(\mu^*)} \mathcal{O}\left(\sqrt{KT\log(T)}\right),$$

where we used that $\sum_{i:\Delta_i \leq \Delta_i'} \mathbb{E}_s[n_{\tau_{i^*}}(i)] \leq T$ in the last line.

For taking care of the sum over arms satisfying (b), define the "good event" $\mathcal{E} = \{\underline{\mu}_i^t \leq \mu_i \leq \overline{\mu}_i^t \, \forall i \in [K] \, \forall t \in [T]\}$. We know that $\mathcal{E}$ occurs with probability at least $1 - 1/T^2$ for any $s \in [0,1]^K$ by merit of Hoeffding's inequality. Under $\mathcal{E}$, we obtain from the standard UCB argument for all $t \leq \tau_{i^*}$ that

$$\mu_{i_t} + 2\sqrt{\frac{2\log(T)}{m_t(i_t)}} \geq \overline{\mu}_{i_t}^t \geq \overline{\mu}_{i^*}^t \geq \mu_{i^*}.$$

This implies that $\Delta_i \leq 2\sqrt{\frac{2\log(T)}{m_{\tau_{i^*}}(i)}}$. Hence, $i \in [K]$ with $\Delta_i > 0$ we get that $m_{\tau_{i^*}}(i) \leq \frac{c\log(T)}{\Delta_i^2}$. Now, post-click rewards are observed for arm $i$ with probability $s_i$ every time $i$ is played by UCB-S, which tells us that $\mathbb{E}_s[m_{\tau_{i^*}}(i)] = \mathbb{E}_s[n_{\tau_{i^*}}(i)]s_i$. It follows from Theorem 5.2 that

$$\sum_{i:\Delta_i > \Delta_i'} \mathbb{E}_s[n_{\tau_{i^*}}(i)]\big(u(s^*, \mu^*) - u(s_i, \mu_i)\big) \leq \sum_{i:\Delta_i > \Delta_i'} \mathbb{E}_s[n_{\tau_{i^*}}(i)] \frac{u(s^*, \mu^*) - u(s_i, \mu_i)}{s_i}$$

$$\leq L \sum_{i:\Delta_i > \Delta_i'} \mathbb{E}_s[n_{\tau_{i^*}}(i)] \frac{|s^*(\mu^*) - s_i| + |\mu^* - \mu_i|}{s_i}$$

$$\leq L \sum_{i:\Delta_i > \Delta_i'} \mathbb{E}_s[n_{\tau_{i^*}}(i)] \frac{|s^*(\mu^*) - s^*(\mu_i)| + \mathcal{O}(H\Delta_i) + \Delta_i}{s_i}$$

$$\leq L \sum_{i:\Delta_i > \Delta_i'} c\log(T) \frac{H\Delta_i + \mathcal{O}(H\Delta_i) + \Delta_i}{s_i\Delta_i^2}$$

$$\leq LH \sum_{i:\Delta_i > \Delta_i'} \mathcal{O}\left(\frac{\log(T)}{s_i\Delta_i}\right)$$

$$\leq LH \sum_{i:\Delta_i > \Delta_i'} \mathcal{O}\left(\frac{\log(T)}{s^*(\mu_i)\Delta_i}\right),$$

where the last line used that $s_i \geq s^*(\mu_i)$ for all $s_i \in \mathrm{supp}(\sigma_i)$ shown in Lemma C.1. This completes the proof of Theorem 5.3.

$\square$

**Lemma D.1.** *Let $i^* \in [K]$ with $\Delta_{i^*} = 0$. For all $\delta > 0$:*

$$\mathbb{P}_{\boldsymbol{\sigma}}\Big(\tau_{i^*} > T - \frac{\sqrt{KT}}{1-\delta}\Big) > 1 - \delta.$$

*Proof.* Suppose the contrary is true, i.e., $\mathbb{P}_{\boldsymbol{\sigma}}\Big(\tau_{i^*} > T - \frac{\sqrt{KT}}{1-\delta}\Big) \leq \delta$. Since $\tau_{i^*} \leq T$ by definition, this implies that

$$\mathbb{E}_{\boldsymbol{\sigma}}[\tau_{i^*}] \leq \delta \cdot T + (1-\delta)\Big(T - \frac{\sqrt{KT}}{1-\delta}\Big) = T - \sqrt{KT}. \tag{14}$$

Now, let $s_{i^*} \in \mathrm{supp}(\sigma_{i^*})$ with $\mathbb{E}_{(s_{i^*}, \sigma_{-i^*})}[\tau_{i^*}] \leq T - \sqrt{KT}$. Note that such $s_{i^*}$ must exist for (14) to hold. We now show that there exists a strategy $s'_{i^*}$ which is a better response to $\sigma_{-i^*}$ than $s_{i^*}$. To this end, similarly to the proof of Lemma C.4, Lemma C.3 tells us that there exists $s'_{i^*} \in [0,1]$ with

$$s'_{i^*} - s^*(\mu_{i^*}) = \Omega\left(\sqrt{\frac{\log(T)}{\mathbb{E}_{(s_i, \sigma_{-i})}[n_{\tau_i}(i)] + \sqrt{KT}}}\right)$$

such that $\mathbb{E}_{(s'_{i^*}, \sigma_{-i^*})}[\tau_{i^*}] = T - \mathcal{O}(1)$. Moreover, recall from Lemma C.4 that $T - \mathbb{E}_{(s_{i^*}, \sigma_{-i^*})}[\tau] < \mathcal{O}(1)$. Then, using $\frac{x + \frac{y}{K}}{\sqrt{x+y}} \geq \sqrt{x+y} - \frac{y}{\sqrt{x+y}}$, equation (8), and Lemma C.2, we obtain

$$v_{i^*}(s_{i^*}, \sigma_{-i^*}) \leq \mathbb{E}_{(s_{i^*}, \sigma_{-i^*})}[n_{\tau_{i^*}}(i^*)]s_{i^*} + \mathcal{O}(1/K)$$

$$\leq \mathbb{E}_{(s_{i^*}, \sigma_{-i^*})}[n_{\tau_{i^*}}(i^*)]\big(s^*(\mu_{i^*}) + (s_{i^*} - s^*(\mu_{i^*}))\big) + \mathcal{O}(1/K)$$

$$\leq \mathbb{E}_{(s_{i^*}, \sigma_{-i^*})}[n_{\tau_{i^*}}(i^*)]\left(s^*(\mu_{i^*}) + \mathcal{O}\left(\sqrt{\frac{\log(T)}{\mathbb{E}_{(s_{i^*}, \sigma_{-i^*})}[n_{\tau_{i^*}}(i^*)]}}\right)\right)$$

$$\leq \mathbb{E}_{(s_{i^*}, \sigma_{-i^*})}[n_{\tau_{i^*}}(i^*)]s^*(\mu_{i^*}) + \mathcal{O}\left(\sqrt{\log(T)\mathbb{E}_{(s_{i^*}, \sigma_{-i^*})}[n_{\tau_{i^*}}(i^*)]}\right)$$

$$\leq \mathbb{E}_{(s_{i^*}, \sigma_{-i^*})}[n_{\tau_{i^*}}(i^*)]s^*(\mu_{i^*}) + \mathcal{O}\left(\sqrt{\log(T)(\mathbb{E}_{(s_{i^*}, \sigma_{-i^*})}[n_{\tau_{i^*}}(i^*)] + \sqrt{KT})}\right)$$

$$< \left(\mathbb{E}_{(s_{i^*}, \sigma_{-i^*})}[n_{\tau_{i^*}}(i^*)] + \frac{\sqrt{KT}}{K}\right)\left(s^*(\mu_{i^*}) + \Omega\left(\sqrt{\frac{\log(T)}{\mathbb{E}_{(s_i, \sigma_{-i})}[n_{\tau_i}(i)] + \sqrt{KT}}}\right)\right)$$

$$\leq \left(\mathbb{E}_{(s_{i^*}, \sigma_{-i^*})}[n_{\tau_{i^*}}(i^*)] + \frac{\sqrt{KT}}{K}\right)\big(s^*(\mu_{i^*}) + (s'_{i^*} - s^*(\mu_{i^*}))\big)$$

$$\leq \mathbb{E}_{(s'_{i^*}, \sigma_{-i^*})}[n_T(i^*)]s'_{i^*} = v_{i^*}(s'_{i^*}, \sigma_{i^*}).$$

Hence, $v_{i^*}(s_{i^*}, \sigma_{-i^*}) < v_{i^*}(s'_{i^*}, \sigma_{-i^*})$, a contradiction. $\square$

## E  PROOF OF COROLLARY 5.4

***Proof of Corollary 5.4.*** The argument roughly follows the standard way to translate an instance-dependent regret bound in multi-armed bandits to a minimax bound (see, e.g., (Lattimore and Szepesvári, 2020)). However, the difference lies in that we split the arms not according to some fixed gap $\Delta'$, but according to the arm-specific gap

$$\Delta'_i := \sqrt{\frac{K\log(T)}{T\, s^*(\mu_i)^2}},$$

which we already used in the proof of Theorem 5.3. This is necessary due to the guarantees of Theorem 5.2 being gap-dependent.

We begin by recalling from equation (13) in the proof of Theorem 5.3 that

$$\sum_{i:\Delta_i \leq \Delta_i'} \mathbb{E}_{\boldsymbol{s}}[n_{\tau_{i*}}(i)]\big(u(s^*,\mu^*) - u(s_i,\mu_i)\big)$$

$$\leq L(H+2) \sum_{i:\Delta_i \leq \Delta_i'} \mathbb{E}_{\boldsymbol{s}}[n_{\tau_{i*}}(i)]\,\Delta_i' \qquad (15)$$

$$\leq \sum_{i:\Delta_i \leq \Delta_i'} \frac{LH}{s^*(\mu_i)}\mathcal{O}\left(\sqrt{KT\log(T)}\right),$$

where we coarsely upper bounded $\mathbb{E}_{\boldsymbol{s}}[n_{\tau_{i*}}(i)] \leq T$.

For all arms $i$ with $\Delta_i > \Delta_i'$, we also get similarly to the proof of Theorem 5.3:

$$\sum_{i:\Delta_i > \Delta_i'} \mathbb{E}_{\boldsymbol{s}}[n_{\tau_{i*}}(i)]\big(u(s^*,\mu^*) - u(s_i,\mu_i)\big)$$

$$\leq LH \sum_{i:\Delta_i > \Delta_i'} \mathcal{O}\left(\frac{\log(T)}{s^*(\mu_i)\Delta_i'}\right)$$

$$\leq LH \sum_{i:\Delta_i > \Delta_i'} \mathcal{O}\left(\sqrt{\frac{T\log(T)}{K}}\right) \qquad (16)$$

$$\leq \sum_{i:\Delta_i > \Delta_i'} \frac{LH}{s^*(\mu_i)}\mathcal{O}\left(\sqrt{KT\log(T)}\right),$$

where we used a very coarse upper bound in the last line by simply adding a factor of $K/s^*(\mu_i)$. Note that the bound in the second last line is a much stronger bound than the one claimed in Corollary 5.4. Combining these two bounds yields the first statement of the corollary.

Recall the definition of $s_{\min} := \min_{i \in [K]} s^*(\mu_i)$ and note that

$$\sqrt{\frac{K\log(T)}{T\,s_{\min}^2}} = \max_{i \in [K]} \Delta_i'. \qquad (17)$$

To get the more refined bound in Corollary 5.4, we can continue from equation (15) and bound the right hand side via a maximum using (17) to get

$$L(H+2) \sum_{i:\Delta_i \leq \Delta_i'} \mathbb{E}_{\boldsymbol{s}}[n_{\tau_{i*}}(i)]\Delta_i' \leq \frac{LH}{s_{\min}}\mathcal{O}\left(\sqrt{KT\log(T)}\right).$$

Lastly, note that in view of equation 16, we have

$$\sum_{i:\Delta_i > \Delta_i'} \mathbb{E}_{\boldsymbol{s}}[n_{\tau_{i*}}(i)]\big(u(s^*,\mu^*) - u(s_i,\mu_i)\big) \leq LH\mathcal{O}\left(\sqrt{KT\log(T)}\right) \leq \frac{LH}{s_{\min}}\mathcal{O}\left(\sqrt{KT\log(T)}\right).$$

The corollary then follows from the regret decomposition in equation (12)

$\square$

## F   PROOF OF THEOREM 5.5

***Proof of Theorem 5.5.*** We work under the utility function $u(s,\mu) = s\mu$. In the strategic click-bandit model there are two distributions associated with each arm, the click distribution $P_{s_i} = \mathrm{Bern}(s_i)$ and the reward distribution $P_{\mu_i}$ with mean $\mu_i$. We here assume that arm $i$'s reward distribution is Bernoulli with mean $\mu_i \in [0,1]$. For convenience, w.l.o.g. we assume that the learner observes both, the click-event and the post-click reward every round. This clearly makes the learning

problem easier for the learner. To summarise the distributions of arm $i$ we let $P_{s_i,\mu_i} = P_{s_i} \times P_{\mu_i}$ denote the product distribution.

We consider problem instances

$$\boldsymbol{\mu} = \left(\frac{1}{2}, \dots, \frac{1}{2}, \frac{1}{2} + \Delta, \frac{1}{2}, \dots, \frac{1}{2}\right)$$

with $\mu_{i^*} = \frac{1}{2} + \Delta$. For convenience, we assume that $M$ is index-independent, i.e., if arm $i$ and arm $j$ have identical distributions $P_{s_i,\mu_i} = P_{s_j,\mu_j}$, then $(n_T(i), n_T(j))$ and $(n_T(j), n_T(i))$ have the same distribution. If $M$ is not index-independent, we can consider different indices $i^*$ for the maximal element in $\boldsymbol{\mu}$. Let us choose $\Delta = c\sqrt{K/T}$ for some constant $c > 0$ to be chosen sufficiently small later.

Let us suppose that $M$ is better than the claimed lower bound so that $R_T(M, \boldsymbol{s}, \boldsymbol{\mu}) \leq o(\sqrt{KT})$ for some $\boldsymbol{s} \in \mathrm{NE}(M, \boldsymbol{\mu})$.[12] By choice of $\Delta$ in $\boldsymbol{\mu}$, it then directly follows that $\mathbb{E}_{\boldsymbol{s},\boldsymbol{\mu}}[n_T(i)] \leq o\left(\frac{T}{K}\right)$ for all $i \neq i^*$, otherwise $R_T(M, \boldsymbol{s}, \boldsymbol{\mu}) \geq \Omega(\sqrt{KT})$. Since $\sum_{i \in [K]} \mathbb{E}_{\boldsymbol{s},\boldsymbol{\mu}}[n_T(i)] = T$, this entails $\mathbb{E}_{\boldsymbol{s},\boldsymbol{\mu}}[n_T(i^*)] \geq \Omega\left(\frac{T}{K}\right)$.

We now show that $\mathbb{E}_{\boldsymbol{s},\boldsymbol{\mu}}[n_T(i)] = o\left(\frac{T}{K}\right)$ cannot hold when $\boldsymbol{s}$ is a Nash equilibrium. To this end, consider an alternative strategy $s_i'$. Now, let $s_i' = s_j$ with

$$j = \underset{k \in [K]}{\operatorname{argmax}} \, \mathbb{E}_{(s_k, s_{-i})}[n_T(k)].$$

Generally, we would expect $j = i^*$, however, $j$ could be any other index in $[K]$ (except for $i$ as we see now). Since $\sum_{k \in [K]} \mathbb{E}_{\widetilde{\boldsymbol{s}}}[n_T(k)] = T$ for any $\widetilde{\boldsymbol{s}}$, we get that $\mathbb{E}_{(s_i', s_{-i}),\boldsymbol{\mu}}[n_T(j)] \geq \frac{T}{K}$. If $i = j$, this would be a contradiction to the statement that $\mathbb{E}_{(s_i, s_{-i}),\boldsymbol{\mu}}[n_T(i)] = o\left(\frac{T}{K}\right)$.

If $j \neq i^*$, we find that $\mathrm{KL}(P_{s_j,\mu_j}, P_{s_i',\mu_i}) = 0$, since $i$ and $j$ have identical click and reward distribution. More generally, we obtain from the chain rule that

$$\mathrm{KL}(P_{s_j,\mu_j}, P_{s_i',\mu_i}) = \mathrm{KL}(P_{s_j}, P_{s_i'}) + \mathrm{KL}(P_{\mu_j}, P_{\mu_i}) = \mathrm{KL}(P_{\mu_j}, P_{\mu_i}) \leq 8\Delta^2,$$

where we used that $\mathrm{KL}(P_{\mu_j}, P_{\mu_i}) \leq \mathrm{KL}(P_{\mu_{i^*}}, P_{\mu_i}) = \mathrm{KL}\left(\mathrm{Bern}(\frac{1}{2}), \mathrm{Bern}(\frac{1}{2} + \Delta)\right) \leq 8\Delta^2$ (see, e.g., Theorem 2.4 in Slivkins et al. (2019)). Recall that $\Delta = c\sqrt{K/T}$. For sufficiently small constant $c > 0$, Theorem 3 in (Garivier et al., 2019) then yields that either

$$\mathbb{E}_{(s_i', s_{-i}),\boldsymbol{\mu}}[n_T(i)] \geq \frac{T}{K} \quad \text{or} \quad \mathbb{E}_{(s_i', s_{-i}),\boldsymbol{\mu}}\left[\frac{n_T(i)}{n_T(j)}\right] \geq \frac{1}{2}. \tag{18}$$

Assuming $n_T(j) \geq 1$, using some algebra (Lemma G.2), the latter can be seen to imply that

$$\mathbb{E}_{(s_i', s_{-i}),\boldsymbol{\mu}}[n_T(i)] \geq \frac{1}{2}\mathbb{E}_{(s_i', s_{-i}),\boldsymbol{\mu}}[n_T(j)] \geq \frac{T}{2K},$$

where the last inequality holds due to the choice of $j$. Hence, from equation 18 we obtain that

$$\mathbb{E}_{(s_i', s_{-i}),\boldsymbol{\mu}}[n_T(i)] \geq \frac{T}{2K}.$$

This leads to a contradiction, as $s_i'$ is a better response to $s_{-i}$ than $s_i$. We have thus shown that $R_T(M, s, \boldsymbol{\mu}) = \Omega(\sqrt{KT})$ for any $\boldsymbol{s} \in \mathrm{NE}(M, \boldsymbol{\mu})$.

$\square$

## G  Technical Lemmas

**Lemma G.1.** *Let $\mathbb{P}$ and $\widetilde{\mathbb{P}}$ be two probability measure (and let $\mathbb{E}$ and $\widetilde{\mathbb{E}}$ denote the respective expectations). Suppose that for integer-valued random variables $N, X_1, X_2, \dots$, it holds for all $k \in \mathbb{N}$ and some $i \in \mathbb{N}$:*

$$\mathbb{E}[N] < \widetilde{\mathbb{E}}[N] \quad \text{and} \quad 0 < \mathbb{P}(X_k = i \mid N) \leq \widetilde{\mathbb{P}}(X_k = i \mid N) \ \text{a.s.} \tag{19}$$

---

[12]We consider pure strategy NE here, though, mixed strategies can be handled analogously.

*Then,*

$$\mathbb{E}\left[\sum_{k=1}^{N}\mathbb{1}_{\{X_k=i\}}\right] < \widetilde{\mathbb{E}}\left[\sum_{k=1}^{N}\mathbb{1}_{\{X_k=i\}}\right]. \tag{20}$$

*Proof.* Note that if $N$ and $X_1, X_2, \dots$ were independent and $X_1, X_2, \dots$ i.i.d. this would immediately follow from Wald's lemma.

We prove the lemma via factorization. It holds that

$$\begin{aligned}
\mathbb{E}\left[\sum_{k=1}^{N}\mathbb{1}_{\{X_k=i\}}\right] &= \sum_{n=1}^{\infty}\mathbb{E}\left[\sum_{k=1}^{n}\mathbb{1}_{\{X_k=i\}} \mid N=n\right]\mathbb{P}(N=n) \\
&= \sum_{n=1}^{\infty}\sum_{k=1}^{n}\mathbb{P}(X_k=i \mid N=n)\,\mathbb{P}(N=n) \\
&\leq \sum_{n=1}^{\infty}\sum_{k=1}^{n}\widetilde{\mathbb{P}}(X_k=i \mid N=n)\,\mathbb{P}(N=n) \\
&= \mathbb{E}\left[\sum_{k=1}^{N}\widetilde{\mathbb{P}}(X_k=i \mid N)\right] \\
&< \widetilde{\mathbb{E}}\left[\sum_{k=1}^{N}\widetilde{\mathbb{P}}(X_k=i \mid N)\right] = \widetilde{\mathbb{E}}\left[\sum_{k=1}^{N}\mathbb{1}_{\{X_k=i\}}\right],
\end{aligned}$$

where in the last line we used that $\widetilde{\mathbb{P}}(X_k=i \mid N) > 0$ almost surely. $\square$

**Lemma G.2.** *Let $X$ and $Y$ be two random variables (which are not necessarily independent) and $Y \geq 1$. Suppose that*

$$\mathbb{E}\left[\frac{X}{Y}\right] \geq \frac{1}{2}.$$

*Then,*

$$\mathbb{E}[X] \geq \frac{\mathbb{E}[Y]}{2}.$$

*Proof.* Basic algebra yields that

$$\mathbb{E}\left[\frac{2X}{Y}\right] - 1 = \mathbb{E}\left[\frac{2X}{Y}\right] - \mathbb{E}\left[\frac{Y}{Y}\right] = \mathbb{E}\left[\frac{2X-Y}{Y}\right] \leq \mathbb{E}\left[2X-Y\right].$$

Hence, if $\mathbb{E}\left[\frac{2X}{Y}\right] \geq 1$, it follows that $\mathbb{E}[2X] \geq \mathbb{E}[Y]$.

$\square$

## H    MORE RELATED WORK

In other related work, Ghosh and Hummel (2013); Liu and Ho (2018); Hron et al. (2022); Hu et al. (2023) study incentive design in online recommendation and are interested in incentivizing agents to contribute high-quality content. They differ to our work primarily in that either the strategies are directly observable, or no bandit learning together with incentive design is performed simultaneously. There is also a multitude of additional work on auction-based mechanism design with unknown agent values and bandit feedback (Gatti et al., 2012; Nazerzadeh et al., 2016; Kandasamy et al., 2023, e.g.). Similar to the previously discussed auction design in MABs (Babaioff et al., 2009; Devanur and Kakade, 2009; Babaioff et al., 2015), Gao et al. (2021) study an auction-based combinatorial multi-armed bandit with payments, where each arm can misreport the cost for its selection. Other related areas of research are dynamic mechanism design (Pavan et al., 2014; Bergemann and Välimäki, 2019) as well as online mechanism design (Parkes, 2007).

# I FUTURE WORK

A natural extension to the studied setting would be to assume that CTRs are user-dependent or more generally dependent on contextual information. Another direction would be to consider multi-slot recommendations in which the learner selects a subset of arms every round and the selected arms compete for the click (and our observations are therefore relative). In fact, the case where the learner selects a set of arms and each arm $i$ is clicked with probability $s_i$ independently of the other arms can be handled with exactly the same methods as presented in this paper. More generally, we believe that the idea of introducing a screening rule based on confidences of each arm's strategy can be extended to various settings and many of our techniques reused.

