# OpenReview forum: "Bandits Meet Mechanism Design to Combat Clickbait in Online Recommendation"
_ICLR.cc/2024/Conference — ICLR 2024 spotlight_

### Official Review · Reviewer_KcYc · 2023-10-26

**Soundness:** 2 fair
**Presentation:** 3 good
**Contribution:** 2 fair
**Rating:** 6
**Confidence:** 4

**Summary:**

This paper proposes a stochastic bandit problem that models a practical situation where vendors in recommendation platforms exaggerates the true values of their products (arms) to increase CTR, which is inconsistent with the objective of recommendation platforms to find an arm $i$ that maximizes a learner's utility $u(s_i, \mu_i)$, where $s_i$ stands for CTR and $\mu_i$ stands for the true mean reward.

The paper aims to design an algorithm that minimizes strong and weak strategic regrets in the face of strategic vendor behavior (i.e., vendors are aware of the algorithm and strategically select the arm CTR before running the algorithm).
The paper presents an algorithm UCB-S, which maintains a set of arms that are estimated not ``clickbaiting'' with high confidence, and runs UCB among arms in the set.
Problem-dependent and problem-independent strong strategic regret upper bounds of UCB-S are both proved to be sublinear over time.
A weak strategic minimax regret lower bound is also provided.

**Strengths:**

* The considered problem of clickbait is common and of interest in both industry and academic. The formulation carefully models the learner's utility $u\$ and arm utility $v$, that generalizes a large class of learner's objective.
* It is novel to combine Nash equilibrium and arm strategy selection in bandit literature. I believe that it well captures the real-world situations where clickbaits are under detection.

**Weaknesses:**

* I feel like it would be more proper and much more interesting if the problem could be modeled by adversarial bandit. Current setting assumes that arms decide their own strategies with knowledge of algorithm in advance (which seems strong to me), and the algorithm is a plain UCB estimating fixed $s$ and $\mu$. However, this problem has the intrinsic of an adversarial strategy selection, that can be well captured by adversarial bandit, in which case, arms dynamically select strategies based on history and thus omit the requirement of knowing algorithm in advance. More importantly, adversarial bandit constructs the interaction between arm strategy selection and learner recommendation/elimination, thus captures real trade-off between incentivizing all arms to be truthful and recommending only the best arms, while current model does not.
* There is confusion in algorithm design and many counterintuitive regret analysis (which I have listed in Questions), that make me doubt the soundness of the results.

**Questions:**

Modeling:
* The proposed model assumes that arms are aware of the algorithm that the learner is going to run in advance. Is this realistic in practice? Further, it is assumed that arms will pick CTRs that form a NE in $NE(M)$, according to Theorem 5.2, does it imply that arms have to know $K$ and $T$ in advance? I recommend presenting how to pick CTRs to form a NE in $NE(M)$. If $K$, $T$, and algorithm $M$ are required to form a NE before running $M$, it appears that the assumption is too strong.
* According to the description of Model 1, strategies $\bf{\it{s}}$ are picked in advance and stay fixed when running $M$, then, the goal of $M$ becomes maximizing learner's utility by estimating $\bf{\it{s}}$ and $\bf{\mu}$. In this case, why are we interested in strong and weak strategic regret, as they take into account the distribution of strategy selection while strategies are just unknown constants throughout the execution of $M$?

Algorithm design:
* I did not see the explicit definition of $\hat{\mu}^t_i$. Is it an estimate of $\mu_i$? If yes, then it is non-trivial to explicitly state how to get $\hat{\mu}^t_i$ by observing $c_{t, i_t}$ and $r_{t, i_t}$ at each round, and I wonder if Step 4 in UCB-S maximizing $\bar{\mu}^t_i$ aligns with the goal of maximizing $u(\cdot)$.
* The algorithm UCB-S only estimates strategies and eliminates bad arms that seem like clickbait, while arms are responsible to pick proper strategies that are less likely to get eliminated. Although it might be tricky for arms to pick strategies, it seems to me that the design of UCB-S is straightforward UCB.

Regret analysis:
* In Section 4, it is not surprising that $(s, \mu)-$Oracle has linear weak strategic regret, since weak strategic regret considers the strategy distribution $\bf{\sigma}$ while the oracle considers only a realization of $\bf{\sigma}$ (In fact, UCB-S also considers a realization of $\bf{\sigma}$ instead of $\bf{\sigma}$, so I doubt whether UCB-S achieves sublinear strong strategic regret as stated in Theorem 5.3).
* I cannot go through all the proofs, but I can hardly follow the proof of Proposition 4.1. In the first glance, it seems to me that the proof is self-contradictory: in Case 1, it is stated that $u(s'+\varepsilon, \mu_{i^*})<u^*_{j^*}$ with $\varepsilon>0, s'+\varepsilon\leq 1$ contradicts the continuity of $u(s, \mu)$ in $s$, while the proof of Case 2 is based on the statement $u(s'', \mu_{i^*})<u^*_{j^*}$ with $s''>s'$. Please double check the proof.
* Theorem 5.3 and Corollary 5.4 provide a problem-dependent and a problem-independent regret upper bound respectively. It is counterintuitive that the problem-dependent bound is larger than the problem-independent bound, as it is often the case that problem-dependent bound is smaller since it additionally takes into account distributional information.
* As previously mentioned, UCB-S considers a realization of $\bf{\sigma}$ as a fixed unknown parameter, and performs straightforward UCB, so I feel like the problem-dependent regret upper bound in Theorem 5.3 can be tightened to order $\log T$. It is of interest to provide a problem-dependent lower bound to show the tightness of the upper bound.

---

> ### Author Response · Authors · 2023-11-15
> **Author Response 1**
>
> Thank you for taking the time to read and review our paper. We believe that there might be some misconceptions about the mechanism design and the nature of our strategic bandit problem, which we hope to resolve below.
>
>
> **Adversarial Bandits**: We are not entirely sure that we understand your comment about adversarial bandits correctly. We believe that, while challenging, it will be possible to extend our work to the case where arms dynamically adapt their strategies (click-rates) to the learning algorithm. We took a first step in this direction with our experiments, in which we considered the case of “learning” arms (as modeled by gradient ascent) and showed that UCB-S still effectively incentivizes desirable arm behavior. However, theoretically analyzing such dynamics is out of the scope of this paper. Moreover, our algorithm UCB-S is not plain UCB (which is incentive-unaware), but---despite being simple in its design---addresses the main difficulty in the strategic click-bandit which is to *incentivize desirable NE under uncertainty* while minimizing regret. Moreover, we also want to stress that our work specifically highlights the trade-off between incentivizing *all* arms to be truthful and recommending only the best arms (see Theorem 5.2 and the discussion below the theorem).
>
>
> **Questions:**
>
> *Modeling:*
> 1. The assumption that the learner publicly commits to an algorithm is standard in the mechanism design literature and is giving the arms maximal knowledege which makes the problem only more difficult for the learner. There are various ways the arms could reach a NE. Prior knowledge of the algorithm $M$, $T$ and $\Delta_i$ would for example be sufficient (see footnote 4). The arms could also reach an equilibrium by repeatedly interacting with the mechanism over several epochs like we considered in our experiments.
>
>
> 2. We believe that there might be a misconception about the goals of incentive-aware online learning. The goal of $M$ is *not* to simply minimize regret given a problem instance $s$ and $\mu$. Instead, $M$ must induce desirable equilibria among arms, which thereby allows $M$ to minimize regret. Under a given mechanism $M$ there can be several NE some of which may be more favourable than others. For example, a mechanism $M$ may induce two NE, the first of which is in desirable strategies, e.g., $(s^*(\mu_1), \dots, s^*(\mu_K))$, and the second in undeseriable strategies $(1, \dots, 1)$ (same goes for distributions over such strategy profiles). Under the first, desirable NE the algorithm would be able to achieve low regret, however, under the second, highly undesirable NE the regret would always be large. Hence, the weak regret is small whereas the strong regret is large. Please let us know if this remains unclear.
>
>
> *Algorithm design:*
>
> 1. Since the distribution of $r_{t,i}$ has mean $\mu_{i}$, we can directly estimate $\mu_i$ from the observations of the post-click rewards: $\hat \mu_i^t = \sum_{l=1}^t \frac{1_{\{i_l=i\}} c_{l,i} r_{l,i}}{m_t(i)}$.
>
> 2. UCB-S is an upper confidence bounds-based algorithm just like UCB, but extended by the screening rule. Despite its simplicity, analyzing the dynamics between strategic arms and the learning algorithm (UCB-S) is challenging and the problem of incentivizing arms under (strategy-)uncertainty while minimzing regret has not been studied before. For this reason, we also believe that keeping the mechansim design simple is highly desirable so that the idea and parts of the analysis can more easily be reused and extended.

---

> > ### Author Response · Authors · 2023-11-15
> > **Author Response 2**
> >
> > *Regret analysis:*
> >
> > 1. We are not sure that we fully understand the question. Please note that the $(s, \mu)$-Oracle is the best response to the actual realization of $\sigma$, i.e., $R_{T}((s, \mu)\text{-Oracle}, \sigma) = E_{s \sim \sigma} [R_T((s, \mu)\text{-Oracle}, s)]$ and since $E[\min_x f(x,y)] \leq \min_x E[f(x, y)]$, it is always going to perform better than the best response to a mixed strategy (i.e., something like a $(\sigma, \mu)$-Oracle). We hope that this clarifies things.
> > Regarding UCB-S, note that the regret analysis of UCB-S builds on Theorem 5.2 which characterizes *all arm strategies $s$ in the support of any NE $\sigma$*. Hence, we can ensure that any realization of $\sigma$ is sufficiently close to the desirable arm strategies, which allows us to bound the strong strategic regret of UCB-S.
> >
> > 2. Please note that $u(s'+ \varepsilon, \mu_{i^*}) < \mu_{j^*}^*$ for some $\varepsilon > 0$ does not generally contradict the continuity of $u(s, \mu)$, but only in Case 1. In Case 1, we have $u(s', \mu_{i^*}) > u_{j^*}^*$ so that together with the above we get $u(s' +\varepsilon, \mu_{i^*}) < \mu_{j^*}^* < u(s', \mu_{i^*})$ for all $\varepsilon > 0$ and $s'+\varepsilon \leq 1$. This is a contradiction to the continuity of $u(s, \mu)$ in $s$ if $s' < 1$, and therefore $s' =1$ must hold in Case 1 as claimed. To make the proof easier to parse, we added an additional sentence that explains that the contradiction of continuity that was used stems from $u(s', \mu_{i^*}) > \mu_{j^*}^*$ in Case 1 and not from the statement $u(s'', \mu_{i^*}) < u_{j^*}^*$ for all $s'' > s'$ with $s'' \in [0,1]$, which is always true by definition of $s'$.
> >
> > 3. The regret bounds in Theorem 5.3 (eq. (3)) and Corollary 5.4 are stated order-wise. We can upper bound the second term in equation (3) by an expression of order $\sqrt{KT \log(T)}$. This is then summarized in Corollary 5.4 with the first term in equation (3) to obtain the order $O(\sqrt{KT\log(T)})$ regret bound. In other words, the regret bound in Corollary 5.4 takes form $c_1 \sqrt{KT \log(T)}+ c_2 \sqrt{KT \log(T)}$ for some constants $c_1, c_2 > 0$. Hence, the bound of Corollary 5.4 is not necessarily better than that of Theorem 5.3. Please also note that problem-independent bounds can be better than problem-dependent bounds. For example, when $\Delta_i = o(1/T)$. In such cases, the dependence on problem-dependent quantities such as $\Delta_i$ leads to vacuous regret bounds, which is why obtaining a problem-independent bound is useful.
> >
> >
> > 4. We want to emphasize that UCB-S (and incentive-aware learning in general) does *not* treat the agent strategies $\sigma$ as a fixed parameter, but aims to incentivize desirable choices of $\sigma$. To do so, UCB-S actively incentivizes agents (arms) in contrast to standard UCB, which fails to incentivize desirable NE. It is also important to us to stress that the first term in Theorem 5.3 is *not* due to the “bandit loss”, i.e., the learning of the reward distributions and the selection of suboptimal arms, but instead due to the fact that the strategic arms exploit our uncertainty (see Theorem 5.2), which causes $\sqrt{KT\log(T)}$ regret. Please also refer to our discussion of the regret bound in the paragraph below Corollary 5.4.

---

> > > ### Author Response · Authors · 2023-11-20
> > >
> > > Dear Reviewer KcYc,
> > >
> > > We hope this message finds you well. We understand that the author reviewer discussion is a critical component of the ICLR review process, and we would like to kindly nudge that the rebuttal period is scheduled to **conclude on November 22nd**. Given that there are only 2 days left before the deadline, we would like to call for your attention to our provided responses.
> > >
> > > In your review you mentioned some concerns and questions about our results, which we believe have been clarified in our responses. We would love to hear if any further information or clarification is needed regarding our response. Thank you for your time and attention to this matter! Your efforts in reviewing submissions are deeply appreciated.
> > >
> > > Best,\
> > > Authors

---

> > > ### Comment · Reviewer_KcYc · 2023-11-21
> > > **Rebuttal Reply**
> > >
> > > I appreciate the detailed reponse from the authors. Most of my concerns are now properly addressed and now I have no further questions. It seems that my rating was a bit too harsh due to some misunderstanding of the modeling. According to the authors, although it seems that arms choose their own CTR, it is the policy who implicitly decide the equilibria among arms since it is known that the arms choose CTR via specific pattern, i.e., maximize clicks. This modeling also mitigate my concern about its application in real world.
> > >
> > > On the other hand, for my question 3 in regret analysis, I cannot quite agree that problem-independent bounds can be better than problem-dependent bounds. In your example, it is not comparable between the two bounds if you give a specific order to $\Delta_i$. But now I understand how you obtain the problem-independent bound, and I feel like I did not properly express my question. In fact, you have two bounds in the same order and they might differ in constant coefficient (in your work, problem-independent bound has a larger one). And that is totally acceptable for me.
> > >
> > > Finally, it still seems to me that the policy design is flat since the clicks and post-click rewards are both observable so their estimations are straightforward. The arm elimination mechanism is also common in bandit literature. However, it is interesting to see the combination of MAB with economic concept. Overall, I feel like I can enjoy this paper much more easier after reading the rebuttal, and I can align with other reviewers for a rating of 6.

---

> > > > ### Author Response · Authors · 2023-11-21
> > > > **Thank you**
> > > >
> > > > Thank you for your swift response. We are happy to hear that we could together resolve your concerns and questions. Your time is very much appreciated.

---

### Official Review · Reviewer_byQz · 2023-10-28

**Soundness:** 3 good
**Presentation:** 4 excellent
**Contribution:** 3 good
**Rating:** 8
**Confidence:** 3

**Summary:**

The authors study the problem of combating clickbait via incentive design in a multi-arm bandit framework. The vendors in the system (e.g., content providers in a recommender system) are modeled as strategic arms. Each vendor/arm has a bernoulli reward distribution with fixed mean, and can choose their clickthrough rate. There is a tradeoff in the utility between higher CTRs (which yields higher engagement) and truthfully reporting a ctr that reflects the true reward (disincentivizing clickbait which is modeled as inflating the CTR for low quality content).

The authors show that ignoring vendor incentives can lead to high regret and design an incentive-aware modification of UCB based on thresholding the reported CTRs to derive low-regret guarantees. They characterize the equilibrium strategies of the agents. Finally, they validate the model and techniques via experiments.

**Strengths:**

The model, while certainly quite stylized, is a neat first step in formulating the incentive issues that arise due to clickbait in the language of bandits. The model is motivated well and explained clearly. The writing is crisp and clear and the paper is overall a pleasure to read. Ultimately I think the paper makes a clear contribution to a growing area of interest; the model, while stylized, is important and will pave the way for future work on aligning vendor incentives with the overall health of the ecosystem.

**Weaknesses:**

Why does the strategic agent behind an arm get to choose the CTR? Is that a realistic assumption? For example, how could a video creator on Youtube choose their CTR? They could direct effort into improving their content which could contribute to increasing CTR, but allowing the agent to choose the CTR as a fixed value seems like a fairly unrealistic assumption.

The strategic aspect is interesting, but one limitation of the model is that the agents are in a sense “stuck” at their particular reward mean $\mu_i$. So an agent’s incentive is solely based on strategically choosing a CTR based on the mechanics of the utility function and learning algorithm. But you would expect that realistically an agent might also put in effort into improving $\mu_i$ (e.g., by improving or changing the quality/genre/tone of content they are producing in a recommender system setting). I feel like that’s an important component that would make this model more compelling, and studying the interplay between clickbait actions (strategically choosing $\mu_i$) and quality-improving actions would be very interesting.

I also think some more work and thought needs to be given on the formulation of agent utilities. The present results hold under some assumptions on utilities like Lipschitnzess and monotonicity of maximizers, and the utility is also based on a single-shot report of the strategically-chosen CTR which seems like a limiting assumption since the agents would probably change their behavior across time steps. But these aspects can be a good direction for future work and doesn’t diminish my opinion of the paper too much.

**Questions:**

I would mainly like to hear the authors thoughts on the first two points in the "weaknesses" section. Namely, (1) How would one go about relaxing the assumption that CTR is a chosen quantity? (2) Can the model be extended to a setting with learning agents who have some degree of flexibility in changing the arm's mean reward?

---

> ### Author Response · Authors · 2023-11-15
> **Author Response**
>
> Thank you for taking the time to read and review our paper. You can find our answers to the two questions you highlighted below.
>
> 1. **CTR as a chosen quantity:** We agree that the case of arms freely choosing click-rates might not portray real-world scenarios. However, please note that when arms are limited in their control of the click-rates, we can expect to suffer less regret (as we remove some of their power to strategize). Still, an additional analysis would be required to make this statement precise. As you already hinted at, an interesting way to relax the mentioned assumption would be to assume that arms can *repeatedly* put effort towards changing their CTR, but there is no guarantee that this effort will have an effect every round (thereby limiting the arms' power/control). This would of course require additional modeling assumptions and the mechanism design and analysis would have to be carefully adapted to these changes. However, we believe that such a problem may still be theoretically tractable (depending on the specific modeling assumptions). In any case, this would also be an interesting problem to study empirically.
>
>
> 2. **Arms changing their mean reward:** Thank you for the interesting question which points at some exciting extensions to our work. Your question is also closely related to your other comment about modeling agent utilities. To give the arms more agency, we could, for instance, allow the arms to change their mean rewards each round at a cost. A simple, natural choice could be that the utility of arm $i$ in round $t$ is given by $v_{t,i} = \lambda_1 1_{\{i_t=i\}} c_{t,i} - \lambda_2 |\mu_{t,i}- \mu_{t-1,i}|$, i.e., arm $i$ has utility $\lambda_1$ for a click minus the cost $\lambda_2$ times distance for changing its mean reward.  We naturally expect the analysis of arm behavior to become more complex (arms now take an action every round). Moreover, in this case a detailed characterization of arm behavior could be very intriguing since we would expect an additional dependence on the parameters of the arm utilities (here, $\lambda_1$ and $\lambda_2$). We believe that there are many similar and interesting extensions in this direction, which can build on our work.

---

> > ### Comment · Reviewer_byQz · 2023-11-21
> >
> > Thanks to the authors for their response. While I still think certain parts of the model are (maybe a bit too) stylized, I think it is a solid and neat first step towards a mechanism design formulation of the problem. I'm happy to keep my score.

---

### Official Review · Reviewer_Se7C · 2023-10-29

**Soundness:** 3 good
**Presentation:** 3 good
**Contribution:** 2 fair
**Rating:** 6
**Confidence:** 3

**Summary:**

The authors study a strategic MAB problem named strategic click-bait. The application scenario of this problem is the online recommendation systems where the content creators or item providers can strategically manipulate the click rate of the items to any arbitrary value between 0 and 1, where the platform aims to achieve high click through rate as well as the consistency between an item's click through rate and the expected reward (normalized) of the item. The authors show that there exist multiple equilibria in the system and propose the UCB-S (UCB with screening) algorithm to penalize the agents for choosing click rates higher than the estimated reward, and proved that the algorithm has \Tilde{O}(\sqrt{ KT }) regret bound in every equilibrium, which outperforms incentive unaware MAB algorithms.

**Strengths:**

Despite some discussion on certain details being missing, the overall presentation is good. The problem settings, algorithm, and main results are clearly presented. I haven't checked all the proof details but the theorems follow the intuition.

**Weaknesses:**

1. The problem setting where the arms can arbitrarily manipulate their click rates from 0 to 1 is an unrealistic assumption. In most real-world online recommendation systems, it is impossible for the item to get arbitrarily close to 1 click rate. There will be limits on the number of items that can appear on the webpage and there will be competition among the item providers, either through sponsors or genuinely improving the item's quality. Due to the limit, only some items will get high click rates but not all. I think the strategic MAB problem is more like an information elicitation problem than a strategic recommendation problem.
2. The UCB-S algorithm is also unrealistic since the punishment is too harsh for any commercial platform. Specifically, the agents are permanently eliminated the first time their report doesn't fall into the estimated expected reward range, and such elimination can be false positives, especially in the early stages of the system. Such false positive eliminations can cause severe regression in brand reputation. I suggest the authors propose alternative solutions that may involve some fixed-length cold-start process to reduce the false positive occurrences or penalize the agents in different ways (like eliminating for a certain period of time instead of permanent elimination), and present the regret bound for the alternatives.
3. It is not easy to understand how the strategic setting in this work is different from previous works in strategic MAB, the authors should consider adding a table with each work's setting, assumptions, solution existence/uniqueness, convergence, and regret bounds to help the readers better understand the novelty.
4. Some details should be elaborated
(1). What is the characterization of NE under the incentive unaware platform? Is it the largest expected reward agent always plays 1 and other agents are indifferent about their strategies? If so, how does this correspond to the result in Figure 3?
(2). Why is UCB-S having higher regret in the early stages than UCB in Figure 4? How is the y-axis value computed?
(3). Under what conditions the NE under UCB-S are pure strategy NE/mixed strategy NE?
5. Minor typo: page 4, "(A3) then ensures that the from the perspective" -> "(A3) then ensures that from the perspective"

I'm on the fence and I think my rating is 5.5, exactly on the borderline since the limitations in the problem setting is really making the UCB-S unrealistic in actual production. I think the theoretical results are interesting, but this is conditioned on my limited experience with the MAB literature.

**Questions:**

Besides questions in the weaknesses, I have the following questions:

1. Will adding a fixed-length cold-start process keep the same regret bound and reduce the false elimination rate of truthful agents?
2. In a more realistic setting, the agents will need to compete for high click rates, how would you model the problem in this case?
3. What application scenarios other than online recommendations are suitable for implementing UCB-S?

---

> ### Author Response · Authors · 2023-11-15
> **Author Response**
>
> Thank you for taking the time to read and review our paper. We respond to your comments and questions below.
>
> 1. **Arms ability to control click-rates:** Allowing the arms to arbitrarily select their click-rate results in a worst-case scenario from the perspective of the learner. If the arms had limited power, we would generally incur less regret. However, some adaptations to the mechanism and an additional analysis is required to make this statement precise. The case where each arm can only change their click-rate within a range (smaller than $[0,1]$) is an interesting direction for future work.
>
>
> 2. **Screening rule and false positives:** Please note that “false positives”, i.e., the event that an arm is eliminated at any round even though it has been truthful, occur with probability less than $1/T^2$ (this probability can be arbitrarily reduced by adapting the confidence intervals). Please also refer to the paragraph at the bottom of page 6, where we discuss that any elimination of UCB-S is always justified w.h.p. in the sense that truthful arms are never eliminated w.h.p. We comment on fixed-length cold-start algorithms further below.
>
>
> 3. **Comparison to other work:** Thank you for the suggestion. To the best of our knowledge, our paper is the first to study the situation where the agent strategies (and other parameters) are initially unobserved and must be learned from interaction while simultaneously incentivizing agents under uncertainty. It is therefore difficult to directly compare to other work along aspects like regret, and we found that the comparison of the regret to standard online learning problems such as MABs is the most insightful as this highlights the cost and challenges of mechanism design in online learning (see e.g. the discussion below Corollary 5.4).
>
>
>
> 4.
>     (1). Yes, in Prop. 4.1 (i) all other arms are indifferent about their strategies because they are *never* selected by the $\mu$-Oracle. Hence, $v_i(s_i, s_{-i}) = 0 \cdot s_i = 0$ for any strategy $s_i$. However, when an arm is selected at least one time (which is the case in Figure 3 since standard UCB explores all arms), then the arm’s utility $v_i$ is maximized by $s_i = 1$.
>
>
>     (2). The observation that UCB-S initially suffer larger regret than UCB can be explained by the elimination rule causing UCB-S to select arms uniformly at random when arms are untruthful. While generating larger regret in the short term (i.e., early epochs), this threat of elimination incentivizes the arms to adapt their strategies in the next epoch and eventually leads to smaller regret for UCB-S. In contrast, UCB simply attempts to minimize regret in the given problem instance without incentivizing any change in the arm behavior. We added additional explanations about this the experiments section. The y-axis is computed directly from the interactions of UCB-S and UCB with the respective arm strategies in response to these algorithms.
>
>
>     (3). This is an open question, which turned out to be difficult to answer in generality. Instead, we deal with mixed NE by approximately characterized *all* strategies in the support of *any* NE (mixed or pure) in Theorem 5.2.
>
> ---
>
> **Questions:**
>
> 1. Thank you for the interesting question. Such fixed-length cold-start algorithms behave similarly to explore-then-commit algorithms and will therefore suffer worse regret (typically order $T^{2/3}$) than adaptive exploration approaches such as UCB. Moreover, the probability of mistakenly eliminating truthful agents will be the same (i.e., there is no notable advantage in terms of false positives), since we must again rely on confidence bounds (please also see our response to your second question about the probability of an unjustified elimination).
>
>
> 2. One way to model such a setting would be to consider the case where the platform selects a *set* of arms each round and only one of the arms is clicked depending on their strategies. For instance, a natural click-model could be the MNL-model. In this case, the arms’ strategies no longer correspond exactly to their click-rate, but instead determine an arm’s click-rate *relative* to other arms (see also future work discussed in Appendix I).
>
>
> 3. While we chose to study a setting particularly inspired by online recommendation, we can think of our work and findings more abstractly as a setting in which the agents’ strategies (among other parameters) are initially unknown and must be learned from interaction while simultaneously incentivzing agents. The concept of confidence widths governing our mechanism design (as is the case in the screening rule of UCB-S) could prove useful for more scenarios, especially, for those where agent strategies are not directly observable, but must be learned from noisy observations. For example, another application scenario, which we however haven't closely looked into yet, could be repeated auctions with noisy bidding.

---

> > ### Comment · Reviewer_Se7C · 2023-11-16
> > **Thank you for the rebuttal and answers**
> >
> > I appreciate the answers from the authors and I think a 6 instead of 5.5 is my rating. I'm still a bit concerned about how well this simplified modeling method can fit the actual production scenario, but this work can be good to have.

---

> > > ### Author Response · Authors · 2023-11-19
> > >
> > > Thank you. Our work is meant to be a first step towards incentive-aware recommendation systems by considering vendor incentives and combining mechanism design with bandit learning. Due to the complexity of the problem, we first wanted to analyze the problem in a basic model, and more realistic extensions are part of active discussions and future investigations. Thank you for your helpful feedback and suggestions.

---

### Official Review · Reviewer_cU9Z · 2023-10-31

**Soundness:** 3 good
**Presentation:** 3 good
**Contribution:** 3 good
**Rating:** 8
**Confidence:** 4

**Summary:**

The paper defines the strategic click-bandit problem, modeling strategic agents represented by arms in classic MAB setting, who are able to choose a click-rate strategically to maximize the utility of themselves under learner's learning policy. It is proved that classic incentive-unaware learning algorithms will fail in this new setting, and an incentive-aware learning algorithm, UCB-S, which elimates arms based on UCB according to the estimations for optimal click-rates and utilities, is proposed. UCB-S is shown to achieve sublinear regret bound uniformaly in every Nash equilibrium among arms.

**Strengths:**

1. The model of strategic click-bandit quite novel and interesting.
2. The theoretical analysis, including ones for incentive-unaware algorithm, new proposed UCB-S, is concrete.
3. The experiments show the robustness of the new proposed algorithm, given each arm running greedy gradient descent.

**Weaknesses:**

1. The motivation of the propose strategic click-bandit setting, modeling recommandation systems, is not quite convincing. Basically, vendors are willing to choose proper item descritions to improve its click-rate, but they are not able to controll the click rate. Especially in the motivated example of clickbait, such a behavior is definitely harm the click-rate in long time. However, in proposed the bandit setting, each arm chooses the click-rate arbitrarily and such a probability will be fixed forever.
2. The theoretical analysis is based on the assumption that Nash equilibrium can be achieved by arms, which may be not realistic. On the other hand, the setting in the experiments that each arm optimize itself by gradient descend would be more realistic one, but maybe much harder to obtain theoretical result.

**Questions:**

Please comment on Weaknesses.

---

> ### Author Response · Authors · 2023-11-15
> **Author Response**
>
> Thank you for taking the time to read and review our paper.
>
> 1. **Arms ability to control click-rates:** We agree that we study a simplified model in which we assume that each arm can freely choose their click-rate. However, this leads to a worst-case scenario from the perspective of the learner and when the arms are limited in their control over the click-rates we generally expect to incur less regret. Regarding clickbait and how clickbait influences click-rates, note that that the click-rate which the vendors can influence can be thought of as the click-rate of a new user without prior exposure to the specific item, hence, the click-rate wouldn't be necessarily harmed over time. Nevertheless, the extension to click-rates which change over time (whether caused by the arms or other sources) is an interesting direction for future work. The case where the arms can only change their click-rate within a limited range is also interesting.
>
> 2. **NE and other arm behavior:** We completely agree. For this reason we wanted to support the developed theory with our experiments which assume a natural model of gradually adapting arm behavior as modeled by gradient descent. We were happy to observe that our incentive design is also effective under such models of arm behavior and that our theoretical results are direclty corroborated by the experiments (e.g., in the experiments we also observe that the optimality gap of each arm governs their behavior/strategy). While challenging, theoretically analyzing such "learning" agents' behavior is definitely an interesting direction for future work but out of the scope of this paper.

---

> > ### Comment · Reviewer_cU9Z · 2023-11-21
> >
> > Thanks for your response.

---

### Meta-Review · Area_Chair_Kp5y · 2023-12-04

**Metareview:**

This paper considers a bandit setting where the arms can be strategic in their report (sort of similarly to an older paper of Braverman and al.). This is motivated by recommender systems, where some sellers/sites can try to manipulate the platform to be displayed more often.

The platform, on the other hand, tries to optimize its cumulative reward.

The setting is obviously a bit simple and not adapted to a direct implementation in real life scenarios. This said, the question of strategic bandits is, I believe (and the reviewers agree) quite important and should be studied in more depth. This paper could set a new trend for instance.

The contributions in themselves are not really technically surprising, but the setting and the concepts introduced are worth being advertised.

**Justification For Why Not Higher Score:**

The results and techniques are not especially new or insightful

**Justification For Why Not Lower Score:**

The model is actually not studied that often (and it should be)

---

### Decision · Program_Chairs · 2024-01-16

Accept (spotlight)